# CDK1-mediated phosphorylation of LDHA fuels mitosis through LDHB-dependent lactate oxidation

Mengting Liu [1,8], Aoxing Cheng [1,8], Weiyi You[2], Jiaxin Wu [3], Chenxu Dai[2], Ting Wang[2], Ying Wu[2], Fumei Zhong[3], Jue Shi [4], Yingying Du[5], Zhonghuai Hou[3], Ping Gao [6], Ke Ruan [3], Yi Yang[7], Yuzheng Zhao [7✉], Kaiguang Zhang [1✉], Zhenye Yang [1,2✉] & Jing Guo [1✉]

## Abstract

While cancer cells overexpress lactate dehydrogenase A (LDHA) to support glycolytic flux and lactate production, the role of LDHB— which preferentially catalyzes lactate oxidation—remains unclear. Here, we demonstrate that LDHB, but not LDHA, is essential for mitotic progression in cancers. During mitosis, CDK1 phosphorylates LDHA at threonine 18, reducing its incorporation into the lactate dehydrogenase (LDH) tetramer. This results in LDHB-enriched tetramers that shift catalytic activity toward lactate oxidation, converting lactate and NAD$^+$ into pyruvate and NADH. The generated NADH fuels oxidative phosphorylation and ATP production, thereby sustaining mitosis. Notably, LDHA-T18 phosphorylation occurs exclusively in tumor tissues. Our findings reveal a tumor-specific mechanism in which CDK1 reprograms LDH isoenzyme composition to direct lactate toward NADH production, ensuring energy homeostasis during mitosis. This underscores the therapeutic necessity of targeting both LDHA and LDHB in cancer.

**Keywords** Lactate; Mitosis; Lactate Dehydrogenase; NADH; ATP
**Subject Categories** Cancer; Metabolism; Signal Transduction

## Introduction

The upregulation of aerobic glycolysis, known as the Warburg effect, is a hallmark of cancer cells that supports their rapid proliferation (Hay, 2016; Vander Heiden et al, 2009). Unlike non-proliferating cells that primarily channel glycolysis-derived pyruvate into mitochondria for ATP production, cancer cells convert a significant portion of pyruvate to lactate. This process regenerates NAD$^+$, sustaining high glycolytic flux to provide both energy and biosynthetic precursors essential for tumor growth. Given that the relative demand for ATP versus biosynthetic building blocks fluctuates throughout the cell cycle, metabolic pathways must be precisely coordinated with cell cycle progression (Icard et al, 2019; Icard and Simula, 2022; Ma et al, 2017). While reciprocal regulation between metabolism and the cell cycle during interphase is increasingly recognized, the metabolic adaptations specifically required during mitosis, a phase characterized by higher demands and productions for energy (Lee and Finkel, 2013; Salazar-Roa and Malumbres, 2017; Wang et al, 2014; Zhao et al, 2019), remain poorly understood.

Mitotic ATP is crucial for powering extensive morphological changes, cytoskeletal reorganization, and organelle dynamics (Salazar-Roa and Malumbres, 2017; Wang et al, 2014; Zhao et al, 2024; Zhao et al, 2019). Since the fate of pyruvate (reduction to lactate vs. mitochondrial oxidation) dictates whether glycolysis fuels energy production or biomass accumulation, its regulation is likely distinct in mitosis compared with interphase. Lactate dehydrogenase (LDH) catalyzes the interconversion of pyruvate and lactate. LDH functions as tetramers composed of varying combinations of LDHA and LDHB subunits (Claps et al, 2022). LDHA preferentially reduces pyruvate to lactate (consuming NADH), an activity upregulated by MYC and hypoxia and associated with tumor aggressiveness (Girgis et al, 2014; Jin et al, 2017; Shim et al, 1997; Zhao et al, 2013). LDHB preferentially oxidizes lactate to pyruvate (producing NADH) and also contributes to tumorigenesis (Brisson et al, 2016; Deng et al, 2022; Hong et al, 2019). The five LDH isoforms (LDH1-LDH5) exhibit different substrate affinities and catalytic preferences based on their LDHA/LDHB composition; for instance, LDH5 (predominantly LDHA) favors lactate production, while LDH1 (predominantly LDHB) favors lactate oxidation (Claps et al, 2022). Although both subunits are frequently overexpressed in cancer (Claps et al, 2022; Wang et al, 2024; Zhang et al, 2024), research has predominantly focused on LDHA, leaving the functional significance of LDHB in cancer metabolism largely unexplored.

[1]Department of Digestive disease, The First Affiliated Hospital of USTC, Division of Life Sciences and Medicine, University of Science and Technology of China, Hefei 230027, China. [2]State Key Laboratory of Immune Response and Immunotherapy, Division of Life Sciences and Medicine, University of Science and Technology of China, Hefei, China. [3]Department of Chemical Physics & Hefei National Research Center for Physical Sciences at the Microscale, University of Science and Technology of China, Hefei, Anhui 230026, China. [4]Center for Quantitative Systems Biology, Department of Physics, Hong Kong Baptist University, Hong Kong, SAR, China. [5]Department of Oncology, the First Affiliated Hospital of Anhui Medical University, Hefei 230022, China. [6]Medical Research Institute, Guangdong Provincial People's Hospital, Guangdong Academy of Medical Sciences, Southern Medical University, Guangzhou, China. [7]Optogenetics & Synthetic Biology Interdisciplinary Research Center, State Key Laboratory of Bioreactor Engineering, Shanghai Frontiers Science Center of Optogenetic Techniques for Cell Metabolism, School of Pharmacy, East China University of Science and Technology, 130 Mei Long Road, Shanghai 200237, China. [8]These authors contributed equally: Mengting Liu, Aoxing Cheng. ✉E-mail: yuzhengzhao@ecust.edu.cn; zhangkaiguang@ustc.edu.cn; zhenye@ustc.edu.cn; jguo2013@ustc.edu.cn

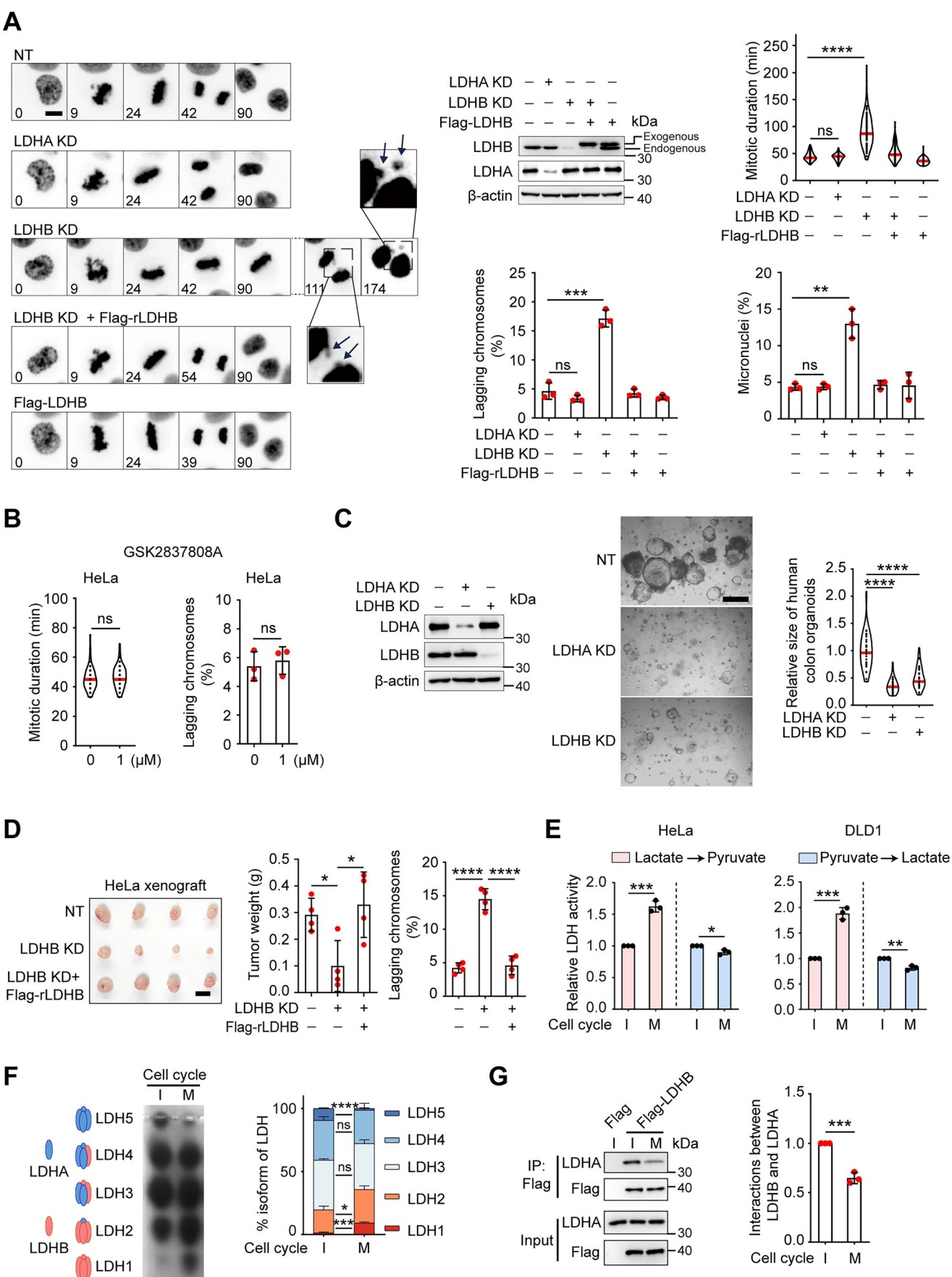

**Figure 1.  LDHB, not LDHA, is required for proper chromosome segregation.**

(A) Endogenous LDHA and LDHB were respectively knocked down (KD) in HeLa cells, and shRNA-resistant Flag-LDHB^WT was expressed as described. Representative mitotic phenotypes of each group are presented. H2B–GFP-labeled chromosomes were monitored by time-lapse microscopy. Scale bar: 10 μm (left). The mitotic duration (min) starting from nuclear envelope breakdown to anaphase onset was quantified ($n = 115, 121, 106, 119, 108$ biologically independent cells; (not significant; ns) $P = 0.1993$, ****$P < 0.0001$) (top right). Quantification of lagging chromosomes and micronuclei is shown ($n = 3$ biologically independent experiments for lagging chromosomes and micronuclei; (ns) $P = 0.2248$, ***$P = 0.0005$, (ns) $P = 0.9121$, **$P = 0.0019$, from left to right) (bottom). NT (non-targeting). (B) HeLa cells (H2B–GFP-labeled) treated with or without LDHA inhibitor GSK2837808A were monitored by time-lapse microscopy. The mitotic duration (min) starting from nuclear envelope breakdown to anaphase onset was quantified ($n = 92, 94$ biologically independent cells; (ns) $P = 0.1739$). Quantification of lagging chromosomes is shown ($n = 3$ biologically independent experiments; (ns) $P = 0.6450$). (C) Human intestinal organoids treated with NT or LDHA/B knockdown were immunoblotted with the indicated antibodies (left). Representative images (middle) and quantitative analysis (right) of each group are presented. Scale bars: 200 μm. ($n = 33, 33, 33$ biologically independent organoids; ****$P < 0.0001$). (D) HeLa cells were subcutaneously inoculated into BALB/c nude mice. Xenograft tumors (left) and tumor weights (middle) at the endpoint were collected and are shown. Scale bar: 1 cm. ($n = 4$ biologically independent tumors for each group; *$P = 0.0147$, *$P = 0.0254$, from left to right). Quantification of lagging chromosomes in xenograft tumor sections are also shown. ($n = 4$ biologically independent tumors for each group, ****$P < 0.0001$) (right). (E) Relative changes in LDH bidirectional activities between interphase (I) and mitosis (M) were quantified in HeLa and DLD1 cells using the enzymatic activity assay kit. ($n = 3$ biologically independent experiments; ***$P = 0.0002$, *$P = 0.0123$, ***$P = 0.0002$, **$P = 0.0022$, from left to right). (F) The functional LDH enzyme is a tetramer composed of varying ratios of LDHA and LDHB subunits. The composition of the five different LDH tetramers is illustrated (left). Interphase and mitotic HeLa cells were collected by double thymidine block and shake-off. LDH isozymes (LDH1-LDH5) were visualized on a gel after native gel electrophoresis (middle). The relative abundance of five distinct LDH tetramer isoforms during interphase and mitosis in HeLa cells is shown ($n = 3$ biologically independent experiments; ****$P < 0.0001$, (ns) $P = 0.2116$, (ns) $P = 0.2116$, *$P = 0.0241$, ***$P = 0.0003$, from top to bottom) (right). (G) HeLa cells were transfected with either Flag-EV or Flag-tagged LDHB. The interaction between endogenous LDHA and exogenous LDHB during interphase and mitosis was confirmed by Co-IP and subsequent immunoblotting with the indicated antibodies (left). Quantified data are presented ($n = 3$ biologically independent experiments; ***$P = 0.0004$) (right). Data Information: Data in (A) (top right), (B) (left), (C) are shown as violin plots. Data were presented as mean ± SD for (A) (bottom), (B) (right), (D–G). $P$ values were determined by unpaired two-tailed Student's $t$-test. Source data are available online for this figure.

In this study, we investigate LDH's role in coordinating metabolism with the cell cycle in cancer cells. We uncover an unexpected, critical function for LDHB-mediated lactate oxidation in maintaining NADH levels during mitosis. Furthermore, we demonstrate that cyclin-dependent kinase 1 (CDK1) phosphorylates LDHA in mitosis, reducing its incorporation into LDH tetramers. This shifts the LDH composition towards tetramers enriched in LDHB, favoring NADH production. The mitotic NADH is subsequently channeled through the malate-aspartate shuttle (MAS) and the glycerol-3-phosphate shuttle (G3PS) to support mitochondrial electron transport, thereby enhancing ATP production. This LDHB-driven metabolic pathway robustly supports mitotic progression and ensures accurate chromosome segregation in cancer cells.

## Results and discussion

### LDHB, not LDHA, is essential for accurate chromosome segregation in cancer cells

To assess the differential roles of LDHA and LDHB in cell cycle progression and tumorigenesis, we individually knocked down LDHA or LDHB in cancer cell lines endogenously expressing both subunits (Figs. 1A and EV1A). Subsequent live-cell imaging comparing knockdown cells with wild-type controls revealed that while depletion of either subunit suppressed cell proliferation (Fig. EV1B), only LDHB knockdown induced prolonged mitosis, lagging chromosomes during mitosis, and increased micronuclei formation after mitotic exit in both HeLa (Fig. 1A) and DLD1 cells (Fig. EV1C). Notably, the majority ($\sim$2/3) of these micronuclei contained centromeres (detected by ACA staining), indicating their origin from lagging chromosomes during mitosis rather than from interphase DNA breaks (Fig. EV1D). To further confirm the specific mitotic defect associated with LDHB loss, we measured the mitotic index in fixed samples. Consistent with the imaging data,

only LDHB knockdown resulted in a significant increase in mitotic index (Fig. EV1E).

To confirm the specificity of the observed mitotic phenotypes to LDHB loss, we treated HeLa and HCT116 cells with GSK2837808A, a selective LDHA inhibitor. Consistent with the knockdown results, GSK2837808A specifically inhibited LDHA activity without impairing mitotic progression (Figs. 1B and EV1F). To extend these findings to more complex models, we investigated the role of LDHB in both patient-derived tumor organoids (Fig. 1C) and xenograft tumors in vivo (Fig. 1D). LDHB depletion significantly inhibited growth in both models. Furthermore, lagging chromosomes were frequently observed specifically in LDHB-knockdown tumors (Fig. 1D). To rule out potential confounding effects from altered lactate metabolism in the organoid model, we measured lactate levels in the culture medium of control organoids over time and found lactate secretion in the medium (Fig. EV1G). Collectively, these data establish that LDHB, but not LDHA, is an essential regulator of mitotic progression and faithful chromosome segregation in cancer cells.

### The shift of the LDH isoenzyme spectrum during mitosis enhances its activity in converting lactate to pyruvate

Given the important role of LDHB in mitosis, we next measured the bidirectional enzymatic activities of LDH tetramers in mitotic vs. interphase cells in both HeLa and DLD1 cells. While the activity of converting pyruvate to lactate (forward reaction) slightly decreased as cells progressed from interphase to mitosis, the activity of converting lactate to pyruvate (reverse reaction) significantly increased in mitosis (Fig. 1E). LDH isoenzyme isoform analysis with native gel electrophoresis revealed that the level of LDH5 (consisting of four LDHA subunits) was notably diminished, while the level of LDH1 (consisting of four LDHB subunits) was significantly increased in the mitotic cells (Fig. 1F). This is likely the result of more LDHB being incorporated into the LDH tetramer during mitosis, and is in line with the increase of reverse reaction in

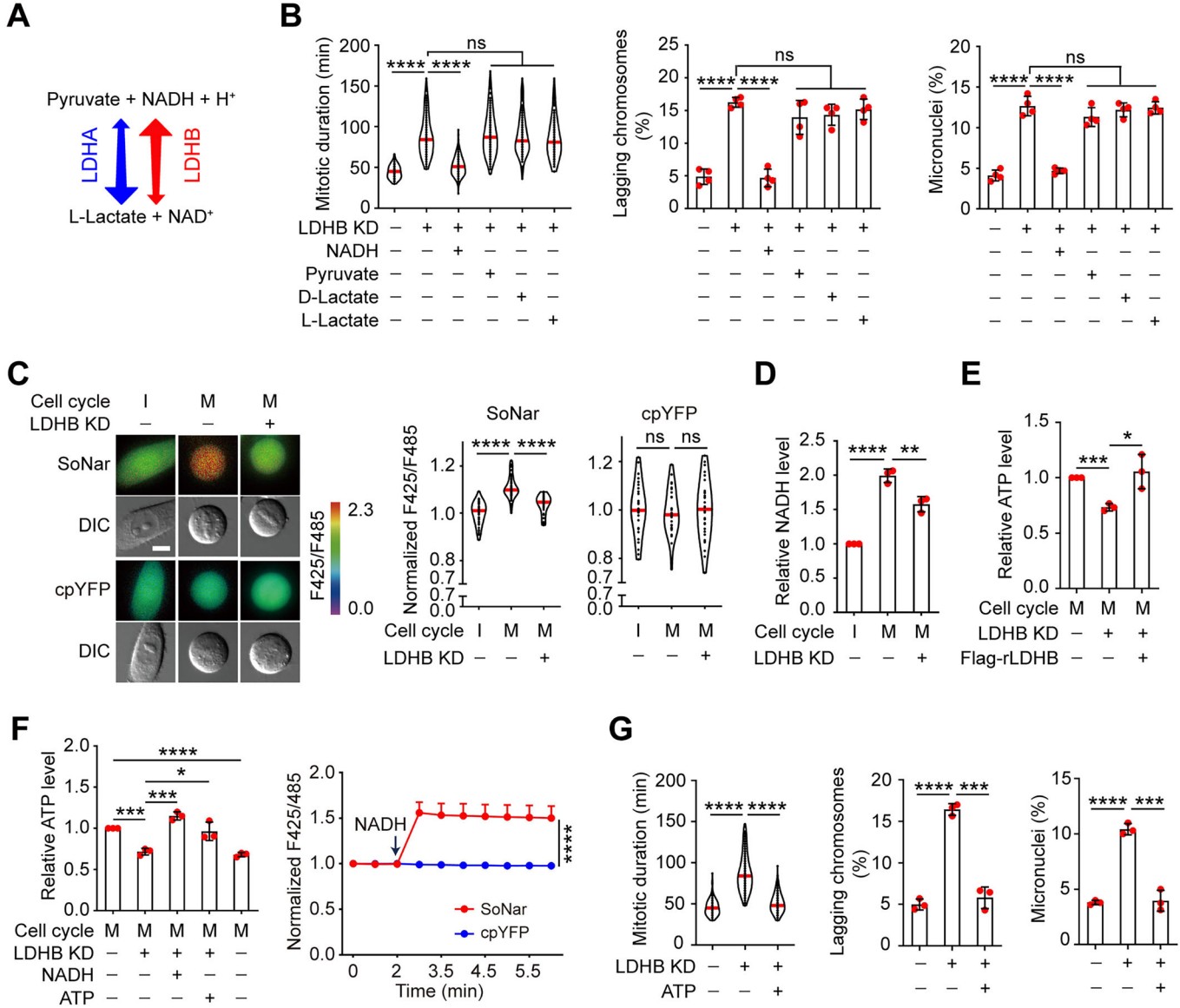

LDHB promotes mitotic progression by enhancing NADH and ATP production

Since LDHB is essential for mitosis and its enzymatic activity increases during this phase, we next investigated whether specific reaction substrates or products of LDHB (Fig. 2A) promote mitotic progression. We first supplemented LDHB-depleted cells individually with D-lactate, L-lactate, pyruvate, or NADH. Live-cell imaging

mitosis. Consistently, immunoprecipitation using Flag-LDHB pulled down less LDHA from the mitotic cells than that from the interphase cells (Fig. 1G). Notably, the expression levels of LDHA and LDHB remain unchanged when cells progress from interphase into mitosis (Fig. EV1H). Thus, these data indicate that LDH tetramers in mitosis comprise more LDHB than those in interphase, and have enhanced enzymatic activity to convert lactate to pyruvate in mitosis.

revealed that only NADH effectively rescued the mitotic defects induced by LDHB knockdown in HeLa/HCT116/DLD1 cells (Figs. 2B and EV2A,B); supplementation with D-lactate, L-lactate, or pyruvate failed to rescue these defects. We then measured the NADH/NAD$^+$ ratio (using the biosensor Sonar) (Zhao et al, 2015), and NADH levels (via biochemical assay) in interphase and mitotic cells. Both the NADH/NAD$^+$ ratio (Fig. 2C) and absolute NADH levels (Fig. 2D) were significantly elevated upon mitotic entry. However, this mitosis-associated NADH increase was markedly reduced following LDHB knockdown (Fig. 2C,D), indicating that LDHB plays a critical role in maintaining elevated NADH levels during mitosis.

Given that NADH fuels the electron transport chain (ETC) in mitochondria for ATP production, the increased NADH resulting from elevated LDHB activity may enhance ATP levels during mitosis. Consistent with this, LDHB knockdown significantly reduced total ATP levels in mitotic cells. This reduction was

◄

**Figure 2.   The LDHB-mediated increases in mitotic NADH and ATP are essential for accurate chromosome segregation.**

(A) Diagram of the bidirectional reactions catalyzed by LDHA and LDHB. (B) In Hela-GFP-H2B cells, LDHB KD cells treated with 1 mM NADH, 1 mM pyruvate, 0.5 mM D-lactate, and 0.5 mM L-lactate were monitored using time-lapse microscopy. The mitotic duration was quantified ($n = 195, 195, 195, 195, 182, 183$ biologically independent cells; ****$P < 0.0001$, (ns) $P = 0.2532$, (ns) $P = 0.9214$, (ns) $P = 0.1645$, from left to right) (left). The quantification of lagging chromosomes is presented ($n = 4$ biologically independent experiments; ****$P < 0.0001$, (ns) $P = 0.1400$, (ns) $P = 0.0765$, (ns) $P = 0.2666$, from left to right) (middle). The quantification of micronuclei is shown ($n = 4$ biologically independent experiments; ****$P < 0.0001$, (ns) $P = 0.1552$, (ns) $P = 0.5410$, (ns) $P = 0.7611$, from left to right) (right). (C) The NADH/NAD$^+$ sensor SoNar and cpYFP (control) were respectively transfected into HeLa cells as described. Cells were monitored by live-cell imaging, and images were pseudo-colored based on the F425/F485 ratio. Scale bar: 10 μm (left). Quantification of intracellular NADH/NAD$^+$ ratio (calculated as F425/F485 ratio) is presented ($n = 53, 53, 53$ biologically independent cells for SoNar; $n = 30, 30, 30$ biologically independent cells for cpYFP; ****$P < 0.0001$, (ns) $P = 0.6581$, (ns) $P = 0.7663$, from left to right) (right). (D) Interphase and mitotic HeLa cells were collected using a double thymidine block and shake-off. Relative changes of intracellular NADH levels in interphase and mitotic cells with or without LDHB KD were measured with an NADH assay kit ($n = 3$ biologically independent experiments; ****$P < 0.0001$, **$P = 0.0083$). (E) Endogenous LDHB was knocked down in HeLa cells, followed by the expression of shRNA-resistant LDHBWT. Mitotic cells were collected as before, and relative changes of intracellular ATP levels were measured using an ATP assay kit ($n = 3$ biologically independent experiments; ***$P = 0.0002$, *$P = 0.0243$). (F) Relative changes of intracellular ATP levels in mitotic HeLa cells upon LDHB KD or NADH/ATP supplement were measured using an ATP assay kit ($n = 3$ biologically independent experiments; ***$P = 0.0003$, ***$P = 0.0003$, *$P = 0.0228$, ****$P < 0.0001$, from left to right) (left). The uptake of NADH was validated using the NADH/NAD$^+$ sensor SoNar and cpYFP (control), and the quantified data are presented ($n = 8$ biologically independent cells for SoNar and $n = 8$ for cpYFP; ****$P < 0.0001$) (right). (G) In Hela-GFP-H2B cells, LDHB KD cells treated with 1 mM ATP were monitored using time-lapse microscopy. The mitotic duration was quantified ($n = 213, 214, 214$ biologically independent cells; ****$P < 0.0001$) (left). The quantification of lagging chromosomes and micronuclei is presented ($n = 3$ biologically independent experiments; ****$P < 0.0001$, ***$P = 0.0002$, ****$P < 0.0001$, ***$P = 0.0005$, from left to right) (right). Data Information: Data in (B) (left), (C), (G) (left) are shown as violin plots. Data were presented as mean ± SD for (B) (middle and right), (D–F), (G) (middle and right). Statistical significance was assessed by two-way ANOVA for (F) (right), and other data were determined by unpaired two-tailed Student's $t$-test. Source data are available online for this figure.

rescued by re-expression of LDHB or supplementation with either NADH or ATP (Figs. 2E,F and EV2C). Notably, the intracellular NADH/NAD$^+$ ratio increased rapidly following extracellular NADH addition (Fig. 2F), although the mechanism of NADH entry remains unclear. Furthermore, ATP supplementation alone rescued both the prolonged mitotic duration and the lagging chromosome phenotype (Figs. 2G and EV2C). Collectively, these findings demonstrate that LDHB-mediated NADH production enhances ATP generation during mitosis.

Cytosolic NADH transfers electrons into mitochondria primarily via the malate-aspartate shuttle (MAS) and the glycerol-3-phosphate shuttle (G3PS) (Hanse et al, 2017; Xiao et al, 2018) (Fig. EV2D). Therefore, we examined whether MDH1 and GPD1L, essential components of MAS and G3PS, respectively, are required for the function of upregulated LDHB activity in mitosis. We generated MDH1 and GPD1L double-knockout (DKO) cells. These DKO cells exhibited an increased incidence of lagging chromosomes, which was not further exacerbated by additional LDHB depletion (Fig. EV2D). Notably, the expression levels of MDH1 and GPD1L remained constant throughout the cell cycle (Fig. EV2E). To distinguish the roles of cytosolic versus mitochondrial NADH in mitotic progression, we supplemented MDH1/GPD1L DKO cells individually with D-lactate, L-lactate, NADH, or ATP. Live-cell imaging showed that only ATP effectively rescued the mitotic defects caused by the DKO (Fig. EV2F), indicating that cytosolic NADH must be shuttled into mitochondria to support mitotic progression.

These data position the mitochondrial NADH shuttles as downstream effectors of LDHB activity, enabling proper chromosome segregation during mitosis. Together, our results demonstrate that LDHB-mediated NADH production fuels the ETC to enhance ATP generation, thereby powering mitotic progression and accurate chromosome segregation.

## Cellular uptake and consumption of lactate increase during mitosis

Given that LDHB strongly catalyzes the conversion of lactate and NAD$^+$ to pyruvate and NADH during mitosis, mitotic cells likely require higher levels of lactate to sustain LDHB activity compared with interphase cells. Indeed, biochemical assays and a newly developed lactate sensor FiLa (Li et al, 2023), revealed significantly elevated intracellular lactate levels in mitotic cells compared with interphase cells (Fig. 3A,B). Since pyruvate conversion to lactate (glycolytic flux) is slightly reduced in mitosis (Fig. 1E), the observed high lactate levels likely originate from extracellular sources. To directly assess lactate uptake, we supplemented the culture medium with lactate and monitored its influx in live cells using the FiLa sensor. We observed a significantly faster increase in intracellular lactate levels in mitotic cells compared with interphase cells, despite their already higher basal lactate level (Fig. 3C). The signal from the control sensor (FiLa C) remained unchanged following lactate addition (Fig. EV3A). Consistent with the rapid lactate influx, the NADH/NAD$^+$ ratio also increased more rapidly in mitotic cells after lactate supplementation (Figs. 3D and EV3B,C).

We further observed elevated MCT1, the primary lactate influx transporter, in mitotic cells compared with interphase or unsynchronized cells, whereas MCT4 expression remained unchanged across cell cycle phases (Fig. 3E). This finding is consistent with a previous study demonstrating progressive MCT1 accumulation throughout the G1, S, G2, and mitosis (Ly et al, 2017), supporting its cell cycle-dependent regulation. Investigating the mechanism underlying increased MCT1 expression during mitosis (at both translational and protein stability levels; Fig. EV3D,E), our data revealed that MCT1 undergoes translational upregulation during mitosis, consistent with previous reports (Stumpf et al, 2013).

To determine if MCT1-dependent lactate uptake promotes mitotic progression, we treated HeLa cells with the selective MCT1 inhibitor AZD3965 (Polanski et al, 2014). While exogenous lactate supplementation induced a twofold increase in intracellular lactate levels during mitosis and accelerated mitotic progression, AZD3965 treatment significantly diminished intracellular lactate uptake and prolonged mitotic duration (Fig. 3F). The promotive effect of lactate on mitotic progression was pronounced in both oxidative tumor cells (SiHa and U-2OS cells, Fig. EV3F) and glycolytic tumor cells (DLD1 and HCT116 cells, Fig. EV3G). Furthermore, lactate supplementation did not influence mitotic progression in cells lacking MCT1 (MDA-MB-231) or LDHB (T47D) (Fig. EV3H).

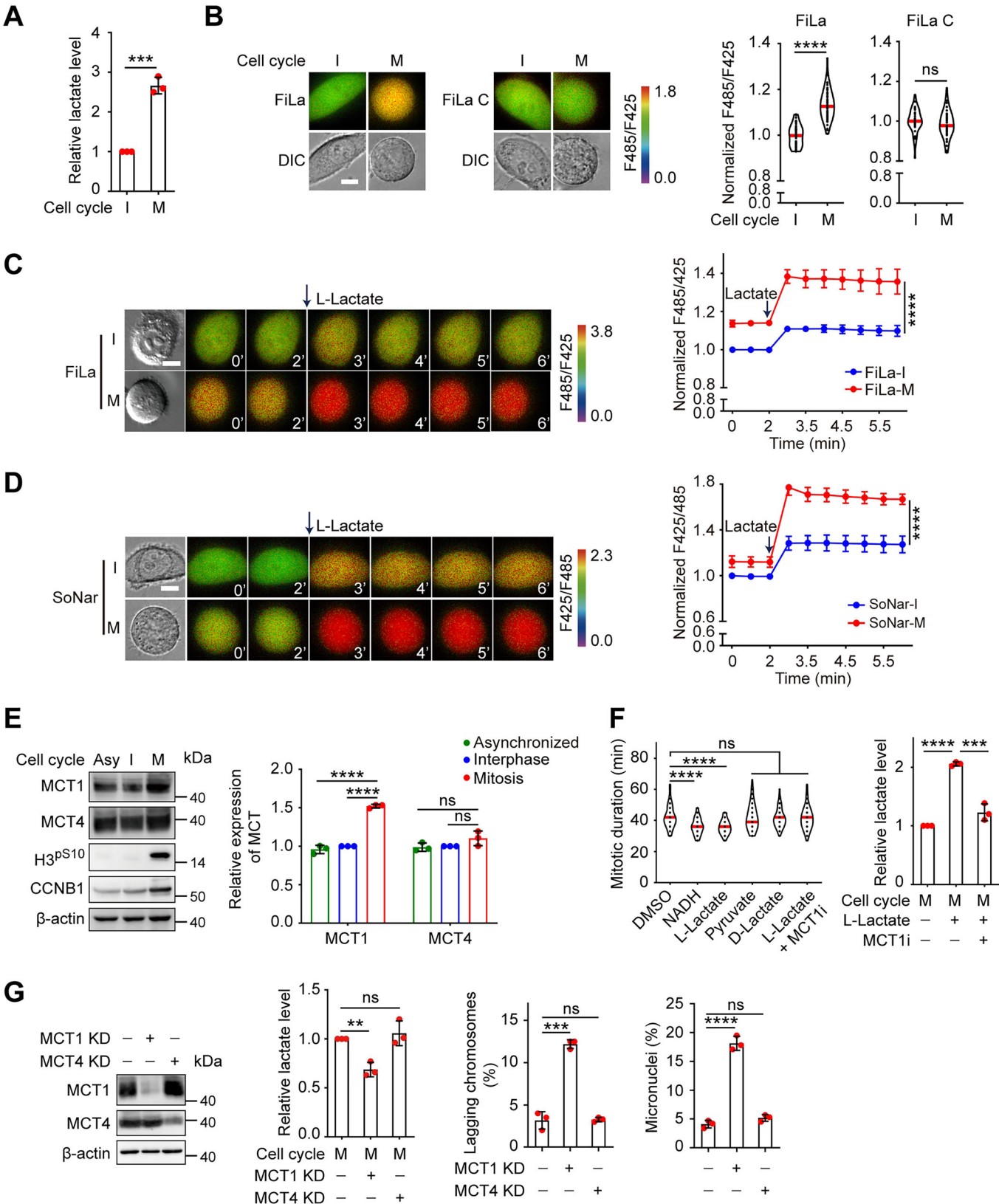

**Figure 3.  MCT1-mediated lactate import increases during mitosis to ensure accurate chromosome segregation.**

(A) Interphase (I) and mitotic (M) HeLa cells were collected using a double thymidine block and shake-off. Relative change of intracellular lactate levels in interphase and mitotic HeLa cells was measured using a lactate assay kit ($n = 3$ biologically independent experiments; ***$P = 0.0002$). (B) The plasmid encoding lactate sensor FiLa and FiLa C (control) were respectively transfected into HeLa cells. Interphase and mitotic HeLa cells were monitored by live-cell imaging and images were pseudo-colored based on the F485/F425 ratio. Scale bar: 10 μm (left). Quantification of intracellular lactate levels (calculated as F485/F425 ratio) is presented ($n = 58$, 58 biologically independent cells for FiLa, $n = 43$, 43 for FiLa C; ****$P < 0.0001$, (ns) $P = 0.2301$) (right). (C) Time-lapse microscopy was performed to monitor the relative changes of intracellular lactate levels using the FiLa sensor in interphase and mitotic HeLa cells, both before and after treatment with 10 mM lactate. Images were pseudo-colored based on the F485/F425 ratio. Scale bar: 10 μm (left). Data were quantified over time ($n = 6$ biologically independent cells for interphase and $n = 6$ for mitosis; ****$P < 0.0001$) (right). (D) Time-lapse microscopy was conducted to measure the relative changes of intracellular NADH/NAD$^+$ ratios using SoNar in interphase and mitotic HeLa cells, both before and after treatment with 10 mM lactate. Images were pseudo-colored based on the F425/F485 ratio. Scale bar: 10 μm (left). Data were quantified over time ($n = 5$ biologically independent cells for interphase and $n = 6$ for mitosis; ****$P < 0.0001$) (right). (E) Western blot analysis was conducted to examine the expression of MCT1 and MCT4 during asynchronized (Asy), interphase (I) and mitotic (M) HeLa cells (left). Cyclin B1 (CCNB1) and H3$^{pS10}$ were utilized as the mitotic markers. The relative expression of MCT was quantified ($n = 3$ biologically independent experiments; ****$P < 0.0001$, (ns) $P = 0.1448$, (ns) $P = 0.1355$, from left to right) (right). (F) Hela-GFP-H2B cells treated with 1 mM NADH, 0.5 mM L-lactate, 1 mM pyruvate, 0.5 mM D-lactate, and 0.5 mM L-lactate in the presence of the MCT1 inhibitor AZD3965 were monitored using time-lapse microscopy. The mitotic duration was quantified ($n = 93$, 92, 93, 92, 91, 93 biologically independent cells; ****$P < 0.0001$, (ns) $P = 0.7535$, (ns) $P = 0.4067$, (ns) $P = 0.3955$, from left to right) (left). The relative changes of intracellular lactate levels in mitotic HeLa cells treated with or without 0.5 mM L-lactate and AZD3965 were measured using a lactate assay kit ($n = 3$ biologically independent experiments; ****$P < 0.0001$, ***$P = 0.0007$) (right). (G) Endogenous MCT1/4 were knocked down in HeLa-GFP-H2B cells, and their expression was examined by western blot (left). Mitotic cells were then collected via double thymidine block and shake-off and subjected to analysis of relative intracellular lactate levels after MCT1/4 KD using a lactate assay kit ($n = 3$ biologically independent experiments; **$P = 0.0018$, (ns) $P = 0.4815$) (middle). MCT1/4 KD cells were imaged using time-lapse microscopy, and the percentage of lagging chromosomes and micronuclei is shown ($n = 3$ biologically independent experiments; ***$P = 0.0002$, (ns) $P = 0.8793$, ****$P < 0.0001$, (ns) $P = 0.0995$, from left to right) (right). Data Information: Data in (B), (F) (left) are shown as violin plots. Data were presented as mean ± SD for (A, C–E), (F) (right), (G). Statistical significance was assessed by two-way ANOVA for (C, D), other data were determined by unpaired two-tailed Student's $t$-test. Source data are available online for this figure.

Moreover, MCT1 knockdown reduced intracellular lactate abundance in mitotic HeLa cells and induced mitotic defects—including lagging chromosomes and micronuclei formation—phenotypically similar to those observed upon LDHB depletion. In contrast, MCT4 knockdown did not induce these defects (Fig. 3G). Collectively, these findings indicate that mitotic cells upregulate MCT1 expression to enhance lactate uptake. This increased lactate availability sustains elevated LDHB activity, driving NADH and subsequent ATP production, thereby promoting mitotic progression and faithful chromosome segregation.

## CDK1-mediated phosphorylation of LDHA at T18 promotes LDHB-enriched tetramers assembly to facilitate lactate consumption and NADH production

We next investigated the mechanism underlying the increased abundance of LDHB-enriched LDH isoforms during mitosis. Given that LDHA and LDHB protein expression remained largely constant throughout the cell cycle (Fig. EV1H), we hypothesized that post-translational modifications by mitotic kinases might regulate their incorporation into the LDH tetramer. Analysis of human serine/threonine phosphoproteomic datasets (Johnson et al, 2023), including mitotic-specific data (Kettenbach et al, 2011; Petrone et al, 2016), revealed that CDK1 potentially phosphorylates LDHA at threonine 18 (T18) within a consensus CDK1 motif (S/T*-P); notably, this motif is absent in LDHB (Fig. EV4A). To validate the functional relevance of T18 phosphorylation, we first performed in vitro kinase assays. Wild-type (WT) LDHA was robustly phosphorylated by CDK1, whereas mutation of T18 to alanine (LDHA$^{T18A}$) completely abolished phosphorylation (Fig. 4A). CDK1 inhibition also blocked LDHA phosphorylation, confirming CDK1 as the responsible kinase (Fig. 4A). In contrast, LDHB was not phosphorylated under identical conditions (Fig. 4A). Functionally, expressing the non-phosphorylatable LDHA$^{T18A}$ mutant attenuated LDH enzymatic activity (lactate-to-pyruvate conversion) in mitotic HeLa cells, while

the phosphomimetic mutant LDHA$^{T18E}$ did not affect activity (Fig. 4B).

We generated a phospho-specific antibody recognizing LDHA pT18 (Figs. 4C,D and EV4B,C). Immunofluorescence (Fig. 4C,D) and immunoblotting (Fig. 4E) confirmed that LDHA T18 phosphorylation occurs specifically in mitotic cells and is absent in interphase. This phosphorylation signal was abolished by CDK1 inhibitors but unaffected by inhibitors of Aurora-A or PLK1 kinases (Fig. 4F). To investigate LDHA$^{T18}$ phosphorylation in more cancer and non-cancer cells, we stained 14 cell lines with LDHA$^{pT18}$ antibody, and found varying levels of phosphorylation in cancer cell lines, with the lowest levels detected in non-cancer cells, suggesting widespread occurrence of mitotic LDHA T18 phosphorylation in cancer (Fig. EV4D).

To determine the impact of T18 phosphorylation on LDHA oligomerization, we performed protein cross-linking assays with purified His-tagged LDHA variants. Western blot analysis revealed that the phosphomimetic LDHA$^{T18E}$ mutant exhibited reduced LDHA-containing tetramer abundance and increased monomer abundance compared with LDHA$^{WT}$ and LDHA$^{T18A}$ (Fig. 4G), indicating that sustained LDHA$^{T18}$ phosphorylation inhibits its tetramer formation. Furthermore, co-immunoprecipitation experiments in mitotic cells showed that the Flag-tagged LDHA$^{T18A}$ mutant exhibited enhanced binding to endogenous LDHA, while LDHA$^{T18E}$ displayed reduced binding (Fig. 4H). These results demonstrate that CDK1-mediated phosphorylation of LDHA T18 during mitosis inhibits the incorporation of LDHA subunits into the LDH tetramer, consistent with the observed decrease in LDH5 levels in mitosis (Fig. 1F). Accordingly, expression of LDHA$^{T18A}$ (but not LDHA$^{T18E}$) induced lagging chromosomes and micronuclei, indicating that T18 phosphorylation is essential for faithful mitotic progression (Fig. 4I).

To elucidate the structural mechanism by which LDHA T18 phosphorylation disrupts tetramerization, we performed molecular dynamics simulations. Analysis of root-mean-square deviation (RMSD) over 250 ns simulations (Fig. EV4E) and monitoring of the

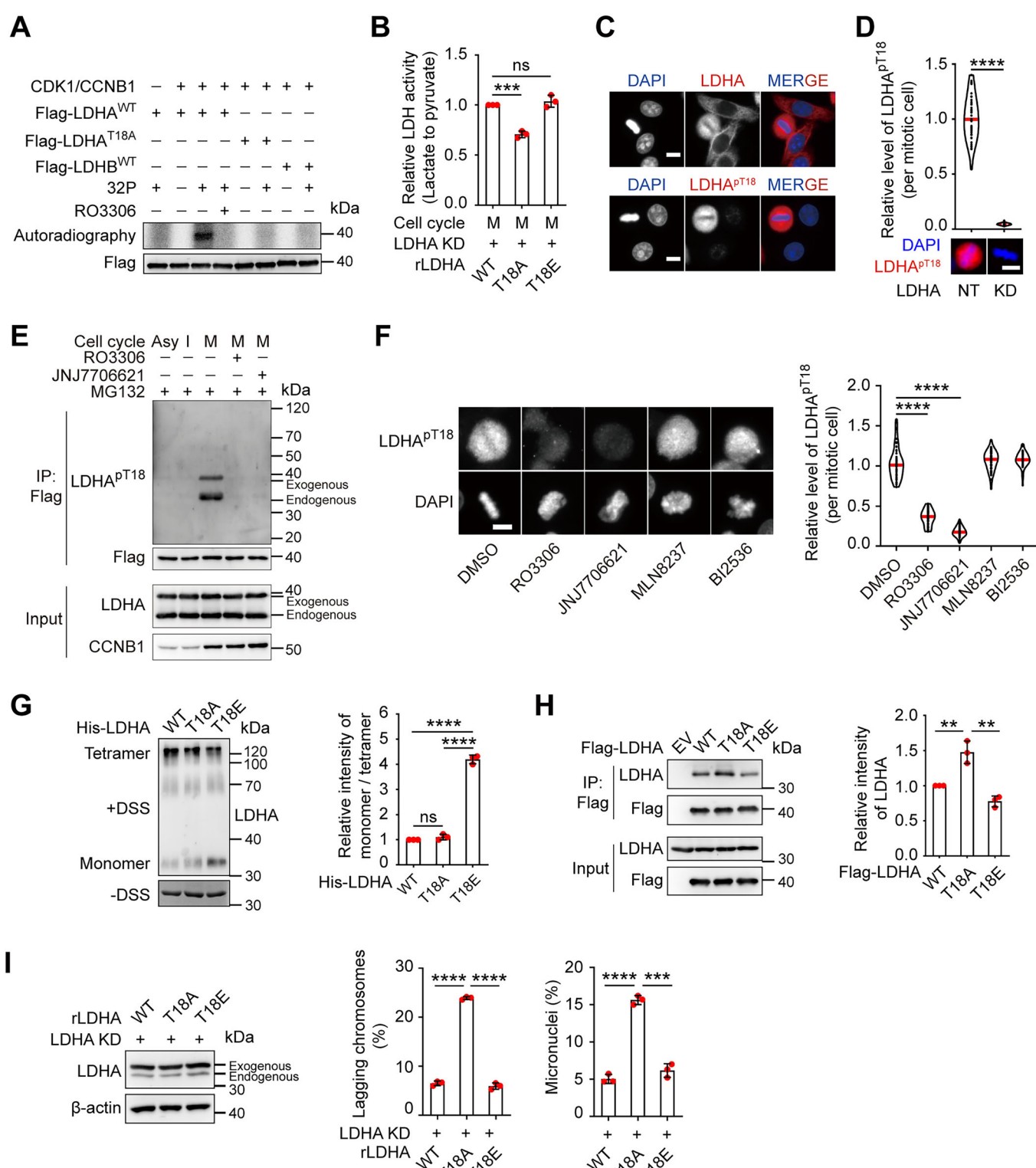

distance between residue T18 (in one monomer) and residue Q297 (in an adjacent monomer) (Fig. EV4E) demonstrated that LDHA T18 phosphorylation dramatically alters the subunit interface conformation. Molecular modeling indicated that phosphorylation destabilizes key interactions at the tetramer interface, involving adjacent residues E16 and Q297 from the neighboring monomer, which are critical for stable tetramer assembly (Fig. EV4F).

### CDK1-mediated phosphorylation of LDHA is essential for mitotic progression and tumor growth

To assess the functional significance of CDK1-mediated LDHA T18 phosphorylation for tumor growth and chromosome segregation in vivo, we generated HeLa cells lacking endogenous LDHA and reconstituted them with either LDHA[WT] or the non-phosphorylatable

**Figure 4. Phosphorylation of LDHA at T18 alters LDH tetramer formation during mitosis.**

(A) Recombinant GST-CDK1 and Flag-LDHA$^{WT}$ or LDHA$^{T18A}$ mutant, Flag-LDHB$^{WT}$ were incubated with $^{32}$P-labeled ATP, followed by SDS-PAGE. The gel was then subjected to autoradiography. (B) Endogenous LDHA was knocked down and shRNA-resistant LDHA$^{WT}$ or mutants were expressed in HeLa cells. The cells were subsequently synchronized in mitosis using a double thymidine block and shake-off method to measure the relative LDH reverse reaction activity via an enzymatic activity kit ($n = 3$ biologically independent experiments; ***$P = 0.0001$, (ns) $P = 0.3310$). (C) Interphase and mitotic HeLa cells were stained with DAPI (blue) and either anti-LDHA or anti-LDHA$^{pT18}$ antibody (red). Scale bar: 10 μm. Immunofluorescence images are representative of three independent experiments with similar results. (D) Relative level of LDHA$^{pT18}$ signal in mitotic HeLa cells upon LDHA KD are shown ($n = 45, 45$ biologically independent cells; ****$P < 0.0001$) (top). Representative immunofluorescence images of LDHA$^{pT18}$ signal are shown. Scale bar, 10 μm (bottom). (E) Exogenous Flag-LDHA$^{WT}$ was expressed in HeLa cells, and then asynchronized (Asy), interphase (I) and mitotic (M) cells treated with or without RO3306 (CDK1 inhibitor), JNJ-7706621 (CDK inhibitor) and MG132 were collected by double thymidine block and shake-off. Flag-tagged LDHA were purified with anti-Flag beads and immunoblotted with the indicated antibodies. (F) Representative images of LDHA$^{pT18}$ signal in mitotic HeLa cells treated with inhibitors against CDK1, CDK, Aurora-A, and PLK1 for 30 min are shown. Scale bar: 10 μm (left). Relative changes of LDHA$^{pT18}$ signal in mitotic HeLa cells following treatment were quantified ($n = 61, 58, 56, 57, 56$ biologically independent cells; ****$P < 0.0001$) (right). (G) His-LDHA$^{WT}$, LDHA$^{T18A}$ or LDHA$^{T18E}$ proteins were purified and treated with or without disuccinimidyl suberate (DSS) for a protein cross-linking assay. The results were analyzed using WB (left), and quantitative data are presented ($n = 3$ biologically independent experiments; (ns) $P = 0.1688$, ****$P < 0.0001$) (right). (H) Exogenous Flag-LDHA$^{WT}$ or T18A/E mutants were expressed in HeLa cells, the interaction between endogenous LDHA and different exogenous LDHA in mitotic HeLa cells was confirmed by Co-IP and subsequent immunoblotting with the indicated antibodies (left). Quantitative data are presented ($n = 3$ biologically independent experiments; **$P = 0.0067$, **$P = 0.0024$, from left to right) (right). (I) In HeLa cells, the endogenous LDHA was knocked down and replaced by shRNA-resistant Flag-LDHA$^{WT}$ or T18A/E mutants. The expressions of LDHA were examined by WB (left). Subsequently, Cells were imaged using time-lapse microscopy, and the percentage of lagging chromosomes and micronuclei is shown ($n = 3$ biologically independent experiments; ****$P < 0.0001$, ***$P = 0.0001$) (right). Data Information: Data in (D, F) are shown as violin plots. Data were presented as mean ± SD for (B, G, H, I). $P$ values were determined by an unpaired two-tailed Student's $t$-test. Source data are available online for this figure.

LDHA$^{T18A}$ mutant. Expression of LDHA$^{T18A}$ significantly impaired tumor cell growth, as evidenced by reduced colony formation in vitro (Fig. 5A) and suppressed growth of xenograft tumors in vivo (Fig. 5B). Furthermore, tumors expressing LDHA$^{T18A}$ exhibited a markedly increased incidence of lagging chromosomes (Fig. 5C), recapitulating the mitotic defects observed in vitro.

To evaluate the clinical relevance of CDK1-mediated LDHA phosphorylation, we examined LDHA T18 phosphorylation levels in paired tumor and paracancerous tissues from colorectal cancer (CRC) patients. Immunofluorescence staining confirmed overexpression of both LDHA and LDHB in CRC tissues compared with adjacent normal mucosa (Fig. EV5A). Critically, phosphorylation of LDHA at T18 was significantly elevated in tumor tissues relative to matched paracancerous tissues (Fig. 5D). These findings indicate that CDK1-mediated LDHA T18 phosphorylation is a prevalent event in actively dividing colorectal tumor cells.

Thus, our study reveals a fundamental cell cycle-coupled metabolic switch: in mitosis, cancer cells suppress lactate generation and instead utilize extracellular lactate as a fuel source. This metabolic rewiring is orchestrated by CDK1-mediated phosphorylation of LDHA at T18, which excludes LDHA from the LDH tetramer, enriches LDHB-high isoforms, and redirects LDH activity towards lactate oxidation. Coupled with MCT1 upregulation, this shift generates NADH and ATP to power chromosome segregation (Fig. 5E).

The Warburg effect, characterized by LDHA-driven lactate production, is a well-established hallmark of cancer proliferation (Chen et al, 2023; Vander Heiden et al, 2009). However, its role during mitosis has remained enigmatic. Our results reveal a mitosis-specific deviation from this paradigm: CDK1-mediated phosphorylation of LDHA at T18 promotes the assembly of LDHB-enriched tetramers, shifting LDH activity toward lactate oxidation. This reconfiguration, together with MCT1 upregulation, increases NADH and ATP availability to support chromosome segregation. Importantly, this metabolic change appears transient, occurring during mitosis and reversing afterward, indicating that cancer cells retain the capacity to switch between lactate production and oxidation depending on the cell cycle phase.

Our findings extend prior observations of metabolic cell cycle coupling by identifying a kinase-dependent switch in LDH isoform composition during mitosis. While consistent with earlier reports that ATP demand peaks during mitotic progression (Pangou et al, 2021; Zhao et al, 2019), they differ from studies that have reported sustained or even increased glycolysis during mitosis in certain contexts (Kang et al, 2020; Maeshima et al, 2018). This discrepancy may reflect differences in cell type, nutrient availability, lactate transporter expression, or methodological approaches to measuring metabolic flux. Future studies using isotope-labeled metabolic tracing in synchronized cells could directly compare lactate flux patterns across multiple tumor models.

Recent work identified a non-metabolic role for lactate in mitosis, where it acts as a signaling molecule stabilizing the APC/C complex via SENP1 inhibition (Liu et al, 2023). Our results suggest an additional, metabolic function for lactate in generating NADH and ATP via LDHB. These two mechanisms are not mutually exclusive and may operate concurrently, with their relative importance influenced by tumor type, metabolic phenotype, or microenvironmental conditions.

Although our data support a model in which LDHB-mediated NADH production fuels the mitochondrial electron transport chain to enhance ATP generation, we did not directly measure mitochondrial NADH or ATP levels during mitosis. It is therefore possible that other metabolic pathways also contribute substantially to mitotic ATP supply. We hypothesize that preferential utilization of lactate-derived NADH by mitochondria during mitosis may be facilitated by cell cycle-dependent modulation of NADH shuttles, but this remains to be experimentally confirmed. The universality of this LDHB–MCT1–ATP axis across tumor types also remains an open question. LDHB is absent or expressed at low levels in some cancers, and LDHA has been reported to catalyze lactate oxidation under certain conditions. Our results, obtained in a subset of glycolytic and oxidative tumor models, suggest that LDHB-driven lactate oxidation is important for mitotic fidelity in these contexts. Broader profiling across cancer types with varied metabolic dependencies will be essential to define the prevalence and functional importance of this mechanism.

From a translational perspective, the ability of CDK1 to reprogram LDH isoenzyme composition suggests potential therapeutic opportunities. Simultaneous targeting of LDHA (to suppress interphase glycolysis) and LDHB (to limit mitotic energy supply), in combination with CDK1 inhibition, could in theory exploit cancer's metabolic vulnerabilities. However, this remains a hypothesis; in vivo validation in additional tumor models, together with toxicity testing in proliferating normal cells, will be necessary to determine whether such strategies are selective and clinically feasible.

In summary, our work proposes a model in which cancer cells dynamically reprogram lactate metabolism: production predominates during interphase to support biosynthesis, while oxidation is favored during mitosis to power chromosome segregation. This phase-specific plasticity, orchestrated by kinase-dependent LDH isoform switching, highlights the sophistication of metabolic regulation during the cell cycle and identifies a potential vulnerability in tumors that depend on lactate oxidation for mitotic progression.

# Methods

## Cell culture and transfection

Cells were cultured in a humidified incubator at 37 °C with 5% $CO_2$ and maintained in Dulbecco's modified Eagle's medium (DMEM, Gibco) supplemented with 5% fetal bovine serum (FBS), 5% newborn calf serum (NCS, Gibco), 100 units/mL penicillin and 100 μg/mL streptomycin (Beyotime Biotechnology, Jiangsu, China). All cell lines were authenticated and tested to be free of mycoplasma contamination. Cell transfection was carried out using polyethylenimine (PEI, Polysciences, Philadelphia, PA, USA).

## Cell synchronization and immunofluorescence

HeLa cells were synchronized using a double thymidine block method. Briefly, exponentially growing HeLa cells were treated with 2 mM thymidine for 19 h, followed by a 9 h release in fresh medium, and then retreated with 2 mM thymidine for an additional 16 h. This procedure resulted in highly synchronized cells at the G1-S phase (Interphase). Subsequently, after a 9 h release in fresh medium, the cells progressed into the M phase. Mitotic cells were then harvested using a mitotic shake-off method (Mitosis). Normally cultured HeLa cells are referred to as unsynchronized cells (Asynchronized).

For Fig. 4E,F, we did the double thymidine treatments and release assay. When cells progressed into mitosis, we treated them with 10 μM MG132 for 30 min, followed by addition of different inhibitors for another 30 min. Finally, the cells were collected for western blot analysis or fixed for immunofluorescence staining.

For DLD1 cells, synchronization was achieved by treating the cells with the CDK1 inhibitor RO3306 at a final concentration of 10 μM for 16 h to arrest them at the G2 phase. After a 1 h release in

**Reagents and tools table**

| Reagent/resource | Reference or source | Identifier or Catalog Number |
|---|---|---|
| **Experimental models** | | |
| Paired tumor and paracancerous tissue sections from colorectal cancer (CRC) patients | The First Affiliated Hospital of Anhui Medical University | N/A |
| BALB/C nude mouse | Shanghai SLAC Laboratory Animal Company | N/A |
| Human colorectal cancer organoids | AIMINGMED | |
| HEK293T | ATCC | CRL-3216 |
| HeLa | CAS, Shanghai | TCHu187 |
| DLD1 | CAS, Shanghai | TCHu134 |
| HCT116 | CAS, Shanghai | TCHu99 |
| MDA-MB-231 | CAS, Shanghai | TCHu227 |
| T47D | CAS, Shanghai | TCHu87 |
| SiHa | CAS, Shanghai | TCHu113 |
| U-2OS | CAS, Shanghai | SCSP-5030 |
| RKO | CAS, Shanghai | TCHu116 |
| SW480 | CAS, Shanghai | TCHu172 |
| MGC803 | FuHeng Biology | FH0269 |
| A549 | CAS, Shanghai | TCHu150 |
| H1299 | CAS, Shanghai | TCHu160 |
| H460 | CAS, Shanghai | TCHu205 |
| SGC7901 | CAS, Shanghai | TCHu46 |

| Reagent/resource | Reference or source | Identifier or Catalog Number |
|---|---|---|
| RPE | ATCC | CRL-4000 |
| MCF10A | CAS, Shanghai | GNHu50 |
| **Recombinant DNA** | | |
| pGIPZ-shLDHB-Flag-rLDHB$^{WT}$ | A Cheng et al, Nature Communications (2019) | N/A |
| His/Flag-LDHA$^{WT}$/LDHB$^{WT}$ | A Cheng et al, Nature Communications (2019) | N/A |
| pGIPZ-shLDHA-Flag-rLDHA$^{WT/}$$^{T18A/T18E}$ | This manuscript | N/A |
| His/Flag-LDHA$^{T18A/T18E}$ | This manuscript | N/A |
| FiLa/FiLa C | Li et al, Cell Metabolism (2023) | N/A |
| SoNar/cpYFP | Zhao et al, Cell Metabolism (2015) | N/A |
| **Antibodies** | | |
| β-actin | Proteintech | Cat# 60008-1-Ig; RRID: AB_2289225 |
| LDHA | Proteintech | Cat# 19987-1-AP; RRID:AB_10646429 |
| LDHA | Cell Signaling Technology | Cat# 3582; RRID:AB_2066887 |
| LDHB | Proteintech | Cat# 19988-1-AP; RRID:AB_10638780 |
| LDHB | Abcam | Cat# ab75167; RRID:AB_1280972 |
| Anti-Centromere Protein Antibody (ACA) | Antibodies Incorporated | Cat# 15-235; RRID:AB_2939059 |
| MDH1 | Proteintech | Cat# 15904-1-AP; RRID:AB_2143279 |
| GPD1L | Proteintech | Cat# 17263-1-AP; RRID:AB_2112359 |
| MCT1 | Millipore | Cat# 2488789 |
| MCT1 | Proteintech | Cat# 20139-1-AP; RRID:AB_2878645 |
| MCT4 | Santa Cruz | K1414 |
| CCNB1 | Proteintech | Cat# 55004-1-AP; RRID:AB_10859790 |
| phospho-H3S10 | Sigma-Aldrich | Cat# 06-570; RRID:AB_310177 |
| phospho-H3S10 | Cell Signaling Technology | Cat# 9706; RRID:AB_331748 |
| FLAG | Transgen Biotech | Cat# HT201; RRID:AB_2927465 |
| phospho-LDHA T18 | GenScript | C438MGASG0 |
| HRP-conjugated anti-rabbit secondary antibody | Jackson ImmunoResearch Labs | Cat# 111-035-003; RRID:AB_2313567 |
| HRP-conjugated anti-mouse secondary antibody | Jackson ImmunoResearch Labs | Cat# 115-035-003; RRID:AB_10015289 |
| Alexa 594-conjugated anti-rabbit secondary antibody | Jackson ImmunoResearch Labs | Cat# 111-585-144; RRID:AB_2307325 |
| Alexa 488-conjugated anti-mouse secondary antibody | Jackson ImmunoResearch Labs | Cat# 115-545-146; RRID:AB_2307324 |
| Alexa 488-conjugated anti-rabbit secondary antibody | Jackson ImmunoResearch Labs | Cat# 111-545-144; RRID:AB_2338052 |

| Reagent/resource | Reference or source | Identifier or Catalog Number |
|---|---|---|
| **Oligonucleotides and other sequence-based reagents** | | |
| His-LDHA-T18A-F | GAAGATCTATGGCAACTCTAAAGGATCAGCTGATTTATAATCTTCTAAAGGAAGAACAGGCC | |
| His-LDHA-T18E-F | GAAGATCTATGGCAACTCTAAAGGATCAGCTGATTTATAATCTTCTAAAGGAAGAACAGGAA | |
| His-LDHA-T18A/E-R | CGGAATTCTTAAAATTGCAGCTCCTT | |
| Flag-LDHA-T18A-F | GAagatctGATGGCAACTCTAAAGGATCAGCTGATTTATAATCTTCTAAAGGAAGAACAGGCC | |
| Flag-LDHA-T18E-F | GAagatctGATGGCAACTCTAAAGGATCAGCTGATTTATAATCTTCTAAAGGAAGAACAGGAA | |
| Flag-LDHA-T18A/E-R | GGggtaccTTAAAATTGCAGCTCCTT | |
| pGIPZ-shLDHA-Flag-rLDHA<sup>WT/T18A/T18E</sup>-F | gcTCTAGAgccaccatggactacaaagACGATGATGACAAAggaagcATGGCAACTCTAAAGGATC | |
| pGIPZ-shLDHA-Flag-rLDHA<sup>WT/T18A/T18E</sup>-R | cgGGATCCCCCAAAGTGTATCTGCACTCT | |
| ShLDHA | CCGGGATCTGTGATTAAAGCAGTAACTCGAGTTACTGCTTTAATCACAGATCTTTTT | |
| ShLDHB | CCGGCGTGATTGGAAGTGGATGTAACTCGAGTTACATCCACTTCCAATCACGTTTTT | |
| Sh MCT1 | CCGGCCAGCGAAGTGTCATGGATATCTCGAGATATCCATGACACTTCGCTGGTTTTTG | |
| Sh MCT4 | CCGGGCTCATCATGCTGAACCGCTACTCGAGTAGCGGTTCAGCATGATGAGCTTTTTG | |
| KO MDH1-F | CACCGCTGGATACTGAGTCGAGGAA | |
| KO MDH1-R | AAACTTCCTCGACTCAGTATCCAGC | |
| KO GPD1L-F | CACCGTGCCTCCACAGTCAAGATGT | |
| KO GPD1L-R | AAACACATCTTGACTGTGGAGGCAC | |
| **Chemicals, enzymes, and other reagents** | | |
| NADH | Sangon Biotech | A600642 |
| Pyruvate | Gibco | 11360–070 |
| Sodium ʟ-lactate | Sigma | 71718 |
| Sodium ᴅ-lactate | Sigma | 71716 |
| 2DG | Sigma | D8375 |
| ATP | Sigma | A2383 |
| Thymidine | Sigma | T1895 |
| GSK 2837808 A | Selleck | #S8590 |
| MG132 | Selleck | #S2619 |
| Cycloheximide (CHX) | Selleck | #S7418 |
| AZD3965 | Selleck | #S7339 |
| RO3306 | Selleck | #S7747 |
| JNJ-7706621 | Selleck | #S1249 |
| MLN8237 | Selleck | #S1133 |
| BI2536 | Selleck | #S1109 |
| DSS | Thermo | 21555 |
| Puromycin | Sigma | P8833 |
| G418 | Sangon Biotech | A600958-0001 |
| NBT | Sangon Biotech | A610379 |
| NAD⁺ | Sangon Biotech | A600641 |
| PMS | Sangon Biotech | A610361 |
| DAKO | Dako | S302380-2 |
| Lipo 3000 | Invitrogen | L3000015 |
| Anti-FLAG M2-agarose | Sigma | M8823 |
| Ni-NTA Agarose | Qiagen | 30210 |

| Reagent/resource | Reference or source | Identifier or Catalog Number |
|---|---|---|
| Recombinant Cdk1-Cyclin B1 complex | Sigma | SRP5009 |
| EASYTIDES ATP, [g-$^{32}$P] | PerkinElmer | NEG502A250U |
| Matrigel | Corning | 356231 |
| **Software** | | |
| ImageJ | ImageJ | https://imagej.net/ |
| GraphPad Prism 8.0 | Graphpad | https://www.graphpad.com/ |
| **Other** | | |
| NAD$^+$/NADH Assay Kit | Abcam | ab65348 |
| EnzyChrom™ ʟ-Lactate Assay Kit | Bioassay systems | ECLC-100 |
| ATP Assay Kit | Beyotime | S0026 |
| QuantiChrom™ Lactate Dehydrogenase Kit | Bioassay systems | D2DH-100 |
| CO$_2$ Independent Medium | Thermo | 18045088 |
| DMEM medium | Gibco | A1443001 |
| MasterAim® Colorectal Cancer Organoid Complete Medium | AIMINGMED | 10-100-018 |
| μ-Slide 8 Well | ibidi | #80826 |
| Glass Bottom Cell Culture dish | NEST | #801001 |
| Four-chamber glass-bottom dishes | Cellvis | D35C4-20-1-N |

fresh medium, the cells entered the M phase, and mitotic cells were harvested using a mitotic shake-off method.

For immunofluorescence analysis, cells were fixed with cold methanol at −20 °C for 5 min. Subsequently, the cells were washed with ethanol and PBS three times, then blocked with 1% BSA in TBST for at least 30 min. The primary antibody was applied for 2 h at room temperature (RT) followed by incubation with the secondary antibody and DAPI staining. Finally, the cells were mounted using DAKO Fluorescence Mounting Medium (Dako North America, S302380-2). Images were captured using a DeltaVision Deconvolution microscope (GE Healthcare, Buckinghamshire, UK) or with Nikon Eclipse Ti-E NIS-Elements AR software (version 4.30.1.10210).

## LDHA knockdown and reintroduction

The pGIPZ-shLDHB-Flag-rLDHB$^{WT}$ plasmid was previously constructed (Cheng et al, 2019). To generate the pGIPZ-shLDHA-Flag-rLDHA$^{WT/T18A/T18E}$ plasmid, the shRNA targeting human LDHB was deleted and replaced with shRNA targeting the 3'UTR of human LDHA. Subsequently, the Flag-tagged human LDHB$^{WT}$ was replaced with Flag-tagged LDHA$^{WT}$, LDHA$^{T18A}$, or LDHA$^{T18E}$ mutant. The primers and shRNA used are provided in the Reagents and Tools Table.

## Construction of knockout mammalian cells

For the generation of CRISPR–Cas9-mediated MDH1/GPD1L double knockout (DKO) cells, we generated PX459 vectors (pSpCas9 (BB)-2A-Puro, Addgene, 48139) containing guide RNA sequences targeting the coding sequences of MDH1 and GPD1L, respectively. The vectors were transfected into cells via Lipofectamine 3000 and subsequently subjected to selection with 1 μg ml$^{-1}$ puromycin. The resulting MDH1/GPD1L DKO clones were analyzed via immunoblotting to confirm the efficiency of gene knockout. The sequences of the guide RNAs used are provided in the Reagents and Tools Table.

## Generation of a phospho-LDHA T18-specific antibody

To investigate the phosphorylation-dependent regulation of LDHA at the T18 site, we developed a site-specific antibody targeting the phosphorylated form of LDHA at T18 (phospho-LDHA T18). The immunogen was a synthetic peptide corresponding to the amino acid sequence surrounding T18, with the threonine residue phosphorylated (pT18). This peptide was conjugated to keyhole limpet hemocyanin (KLH) to enhance immunogenicity. New Zealand rabbits were immunized with the phosphorylated peptide using a standard protocol, including an initial immunization followed by three boosts at 2-week intervals. Rabbit serums were collected at each step and tested for specificity via indirect enzyme-linked immunosorbent assay (ELISA) using phosphorylated and unphosphorylated peptides. Immunofluorescence and western blot analysis were performed to confirm specificity. Highly specific antibodies were subsequently purified using affinity chromatography with the phosphorylated peptide as the immobilized ligand. The purified antibodies were validated through western blot, dot blot and immunofluorescence imaging to ensure

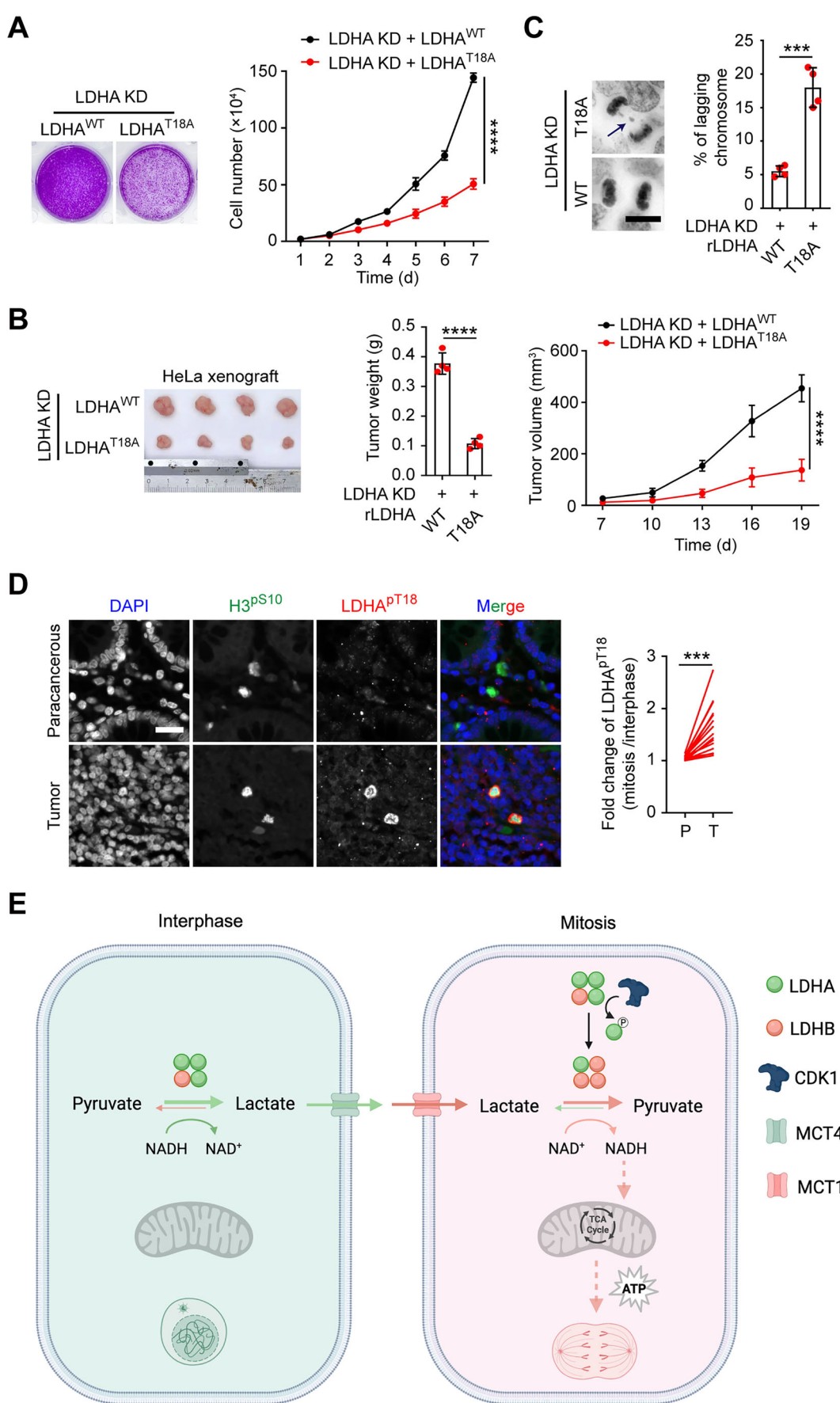

**Figure 5. Phosphorylation of LDHA at T18 is essential for tumor progression.**

(A) In HeLa cells, endogenous LDHA was knocked down, followed by expression of shRNA-resistant LDHA$^{WT}$ or LDHA$^{T18A}$. A colony formation assay was conducted, and cells were fixed after culturing for 10 days. Crystal violet staining of the cells was shown (left). Additionally, a cell proliferation assay was performed with the HeLa cells, and the growth curves were plotted over seven days ($n = 3$ biologically independent samples for each group; ****$P < 0.0001$) (right). (B) Endogenous LDHA was knocked down and followed by expression of shRNA-resistant LDHA$^{WT}$ or LDHA$^{T18A}$ in HeLa cells. HeLa cells were subcutaneously inoculated into nude mice. Xenograft tumors were collected at the endpoint and shown (left). The weight of tumors at the endpoint was analyzed ($n = 4$ biologically independent tumors for each group; ****$P < 0.0001$) (middle). The growth curve of the tumors was measured ($n = 4$ biologically independent mice for each group; ****$P < 0.0001$) (right). (C) Immunofluorescence staining of DAPI in xenograft tumor sections is shown. Scale bar: 10 μm. Quantification of lagging chromosomes in xenograft tumors is presented ($n = 4$ biologically independent tumors for each group; ***$P = 0.0002$). (D) Images of the paracancerous and colorectal cancer (CRC) samples stained with DAPI (blue) and antibodies for LDHA$^{pT18}$ (red) and H3$^{pS10}$ (green) is shown. Scale bar, 20 μm (left). Relative fold change of LDHA$^{pT18}$ (mitosis/interphase) in paired CRC samples ($n = 15$ biologically independent samples; ***$P = 0.0001$), P: paracancerous, T: tumor (right). (E) Working model of CDK1-mediated LDHA phosphorylation fuels mitosis through LDHB-dependent lactate oxidation. Our findings demonstrate that the mitotic kinase CDK1 phosphorylates LDHA T18 during mitosis, which reduces its incorporation into LDH tetramers and promotes the formation of LDHB-enriched tetramers. This modification facilitates increased NADH and ATP production, thereby driving mitotic progression and ensuring precise chromosome segregation (image created with https://biorender.com). Data Information: Data were presented as mean ± SD for (A–C). Data in (D) are shown as symbols and lines. Statistical significance was assessed by two-way ANOVA for (A), (B) (right); by paired two-tailed Student's $t$-test for (D); other data were determined by unpaired two-tailed Student's $t$-test. Source data are available online for this figure.

their ability to specifically recognize the phosphorylated form of LDHA at T18.

## Time-lapse imaging of mitotic progression

Stable cell lines transfected with GFP-H2B were seeded onto an eight-chambered coverglass (μ-Slide 8 Well, ibidi; 80826) for live-cell imaging. The original culture medium was replaced with CO₂-independent medium (Thermo Fisher Scientific, 18045088), which was supplemented with NADH (1 mM), pyruvate (1 mM), ATP (1 mM), D-lactate (0.5 mM), and L-lactate (0.5 mM), or various inhibitors as required. Images were captured every 3 min with an exposure time of 0.2 s for 24 h using a 20× objective lens on an inverted fluorescence microscope (Nikon Eclipse Ti-E). The imaging medium was maintained at 37 °C throughout the experiment. Image sequences were analyzed using ImageJ software, and mitotic phenotypes were manually assessed.

## Cell lysis, immunoprecipitation, immunoblot, and dot blot

Cells transfected with Flag-tagged proteins were lysed using a buffer composed of 50 mM Tris-HCl (pH 7.4), 150 mM NaCl, 1 mM EDTA, 1% Triton X-100, 50 mM NaF, 1 mM Na₄P₂O₇, 1 mM Na₃VO₄, 1 mM PMSF, and a protease inhibitor cocktail (Sigma, P8340). For co-immunoprecipitation (Co-IP) with Flag-tagged proteins, the cell lysates were incubated with anti-Flag M2-Magnetic Beads (Sigma, M8823) for 4 h at 4 °C. The beads were then washed three times with TBS (50 mM Tris-HCl, 150 mM NaCl, pH 7.4), and the Flag-tagged proteins were subsequently eluted using Flag peptides (bank peptide). For Western blot analysis, the immunoprecipitates or whole-cell lysates were boiled at 95 °C for 5 min and separated on 10% SDS-polyacrylamide gels. The separated proteins were then transferred to polyvinylidene difluoride membranes (Millipore, Bedford, MA, USA). The membranes were blocked using 5% non-fat milk in Tris-buffered saline containing 0.1% Tween 20 for 1 h before incubation with primary and secondary antibodies. Detection of signals was performed using the Western Lightning Chemiluminescence Reagent Plus (Advansta, Menlo Park, CA, USA). Quantitative analysis was carried out utilizing the ImageJ software. For the dot blot assay, 1 μL of varying concentrations of LDHA$^{T18}$ non-phosphorylated and phosphorylated peptide solutions were dotted

onto a nitrocellulose membrane and allowed to air-dry completely at room temperature. The membrane was then blocked and incubated with primary and secondary antibodies following the same protocol used for Western blotting.

## LDHB enzymatic activity assay

The activity of LDHB (lactate to pyruvate) was assessed using the QuantiChrom™ Lactate Dehydrogenase Kit (Bioassay Systems, D2DH-100) according to the manufacturer's instructions. LDHB enzyme activities were normalized to protein concentrations, which were determined via the Bradford assay.

## LDHA enzymatic activity assay

HeLa cells in both interphase and mitosis were harvested and lysed using an NP-40 lysis buffer composed of 50 mM Tris-HCl (pH 7.4), 150 mM NaCl, 0.3% Nonidet P-40, supplemented with 50 mM NaF, 1 mM Na₄P₂O₇, 1 mM Na₃VO₄, 1 mM PMSF, and a protease inhibitor cocktail. The activity of LDHA (pyruvate to lactate) was assessed in a reaction buffer containing 20 mM Tris, 50 mM KCl, 1 mM NADH, and 10 mM pyruvate (pH 7.0). The decrease in absorbance at 340 nm, indicative of NADH oxidation, was quantitatively measured using a Microplate reader (Thermo Fisher Scientific).

## LDH isoform characterization

Native gel electrophoresis was employed to separate and characterize the five known lactate dehydrogenase (LDH) isoforms, using methods previously described (Ross et al, 2010). Briefly, HeLa cells in both interphase and mitosis were harvested and lysed in a cold buffer containing 100 mM potassium phosphate (pH 7.0) and 2 mM EDTA. The lysates were then loaded onto a 1.2% agarose gel prepared in 25 mM Tris-HCl and 250 mM glycine (pH 9.5) buffer. A 6× loading buffer consisting of 0.1% bromophenol blue, 15% glycerol, and 20 mM Tris-HCl (pH 8.0) was used. Electrophoresis was performed over 240 min at 110 V in a 5 mM Tris-HCl and 40 mM glycine (pH 9.5) running buffer. Following electrophoresis, the gels were briefly rinsed in 100 mM Tris-HCl (pH 8.5) buffer. LDH isoenzymes were visualized after incubating the gel for 20 min at 37 °C in a staining solution containing lactate (3.24 mg/mL), β-nicotinamide adenine dinucleotide (NAD⁺, 0.3 mg/mL), nitroblue

tetrazolium (NBT, 0.8 mg/mL), and phenazine methosulfate (PMS, 0.167 mg/mL), all dissolved in 10 mM Tris-HCl (pH 8.5) buffer.

## Protein expression, purification, and DSS cross-linking analysis

Recombinant plasmids pET-22b-His-LDHA$^{WT/T18A/T18E}$ were constructed and transformed into BL21 *E. coli*. Transformed bacteria (50 μL) were plated on LB agar plates containing 100 μg/mL ampicillin (amp) and incubated at 37 °C for 12 h. A single colony was inoculated into 50 mL of LB broth supplemented with 100 μg/mL amp and cultured with shaking at 37 °C for 12 h. This was followed by the addition of 125 mL of fresh LB containing amp, and the culture was grown until the optical density (OD) at 600 nm reached 0.5–0.6. To induce protein expression, 125 mL of fresh LB containing amp and 0.5 mM IPTG was added, and the culture was incubated at 16 °C for 18 h. Cells were harvested by centrifugation at 6200 × *g* for 10 min at 4 °C, and the pellet was resuspended in 6 mL of His buffer (20 mM Tris-HCl, 500 mM NaCl, 3% glycerol, pH 7.9) containing 1 mM PMSF. Bacterial lysis was achieved through ultrasonication, followed by centrifugation at 18,000 × *g* for 20 min at 4 °C to collect the supernatant containing His-tagged proteins. The proteins were then bound to an NTA nickel column (Qiagen, Germany, 30210) for 2 h at 4 °C and eluted with 60 μL of 250 mM imidazole. Protein crosslinking with disuccinimidyl suberate (DSS) was conducted as described in the protocol (Liu et al, 2018), where 2 mM DSS in HEPES buffer (40 mM HEPES, pH 7.5, 150 mM NaCl, 0.1% NP-40) was added to the protein supernatants for 30 min at room temperature. Finally, the samples were boiled, resolved by 15% SDS-PAGE, and analyzed by Western blotting using an LDHA-specific antibody.

## Biochemical quantification of L-lactate/NADH/ATP

Intracellular levels of L-lactate, NADH, and ATP were measured using the EnzyChrom™ L-Lactate Assay Kit (Bioassay Systems, ECLC-100), the NAD⁺/NADH Assay Kit (Abcam, ab65348), and the ATP Quantification Kit (Beyotime Biotechnology, S0026), respectively, in accordance with the manufacturers' instructions. The measured values were normalized to protein concentrations, which were determined using the Bradford assay.

## In vitro kinase assay and autoradiography

The in vitro kinase assay was performed in a 20 μL volume containing 1× kinase buffer (50 mM Tris-HCl (pH 7.4), 10 mM MgCl₂, 2 mM DTT), 100 ng of recombinant CDK1/cyclin B1 complex (Sigma, SRP5009), 1 μg of Flag-tagged protein, and 5 μCi of γ-32P ATP (PerkinElmer, NEG502A250U), with the optional inclusion of 10 μM RO3306. Reactions were incubated at 30 °C for 1 h before being terminated with 5× loading buffer. Proteins were then heated at 95 °C for 5 min and separated by 10% SDS-PAGE. Autoradiography to detect radioactivity was conducted using a Typhoon Isotope Laser Confocal Imager (Amersham Typhoon IP).

## Live-cell imaging with biosensors

Plasmids encoding the NADH/NAD⁺ biosensor (SoNar) and the lactate biosensor (FiLa) were respectively transfected into HEK293T cells, using a 2:1:1 ratio with plasmids pMD2.G and psPAX2, to produce lentivirus carrying the biosensor gene. 48 h post-transfection, the virus were harvested, and HeLa cells were transduced with the resultant lentivirus carrying either the NADH/NAD⁺ biosensor or the lactate biosensor. For the detection of NADH/NAD⁺ and lactate in interphase and mitotic HeLa cells, cells were plated on a 35-mm glass-bottom dish with DMEM medium. Images were acquired using DeltaVision softWoRx software (version 6.5.2). To monitor the responses of sensor-expressing cells to lactate addition during interphase and mitosis, live-cell imaging was performed on interphase and mitotic HeLa cells for 2 min in DMEM medium (Gibco, A1443001) supplemented with 10% FBS and 1% P/S. Subsequently, 10 mM lactate was gently added, and images were captured every 30 s for a total of 6 min. Imaging data were analyzed with ImageJ software, and fluorescence ratios were normalized to the interphase group.

## Structure analysis and molecular dynamics simulation

Chain B and chain C of LDH (PDB entry 4ojn) were used to set up molecular dynamics (MD) simulations. The periodic boundary condition (PBC) with a cubic box type was used, with the minimum distance of 1.0 nm between the solute and the box boundary. The box was filled with SPC water molecules. NaCl was used to compensate for the net charge of the solute and make the system neutral. The simulation was conducted by using the leap-frog algorithm (Hockney et al, 1974) with a 1 fs time-step. Covalent bonds in the protein were constrained using the P-LINCS algorithm (Hess, 2008). The cutoff distance for van der Waals interactions were chosen to be 1.2 nm, and the neighbor list was updated every 10 fs. The long-range electrostatic interactions were treated by the PME algorithm, with an interpolation order of 4.

All MD simulations were performed with a parallel implementation of the GROMACS-2024.2 package, using a modified GROMOS 54A8 force field (Margreitter et al, 2013; Margreitter et al, 2017; Petrov et al, 2013). The system with the solute and water was energy-minimized by the steepest descent method. Then we performed a 100-ns equilibration simulation and used its final state structure for the subsequent production simulation. The initial atomic velocities were generated according to a Maxwell distribution at 300 K. The simulation was performed under the constant NPT condition. Two groups (protein and non-protein) were coupled separately to a temperature bath of 300 K by using a velocity rescaling thermostat (Bussi et al, 2007), with a relaxation time of 0.1 ps. The pressure was kept at 1 bar with a relaxation time of 5.0 ps and the compressibility of $4.5 \times 10^{-5}$ bar$^{-1}$. In the production run, LDHA$^{T18}$ or LDHA$^{Y10}$ residue on both chains were phosphorylated, respectively. A 1-μs production run was conducted. We also conducted a 1-μs control run in which the protein was kept wildtype for comparison.

## Cell proliferation, colony formation

HeLa cells were seeded at a density of $2 \times 10^4$ cells per well in triplicate and counted daily over a 7-day period to monitor cell proliferation. For colony formation, $1 \times 10^4$ cells were seeded in six-well culture dishes. After ten days, cells were fixed with 4% PFA and stained with 0.2% crystal violet.

## In vivo xenograft experiments

Five-week-old BALB/c nude mice were purchased from Shanghai SLAC Laboratory Animal Company. Animals were housed within a specific pathogen-free facility, maintained at a temperature range of 20–22 °C. The environment was regulated to ensure a 12-h light cycle and 12-h dark cycle, with relative humidity levels kept between 50 and 60%. All animals had free access to food and water throughout the study. All experimental procedures involving mice were conducted according to the National Guidelines for Animal Usage in Research (China) and were approved by the Animal Research Ethics Committee at the University of Science and Technology of China (permit 2020-N(A)-280). Both male and female mice were randomly assigned to different groups for this study. For xenograft experiments, $5 \times 10^6$ HeLa stable cell lines were collected and resuspended in 50 μL PBS. The suspension was mixed with 50 μL Matrigel and injected subcutaneously into the left or right flank of BALB/c nude mice. Tumor formation was assessed every 2 days, and tumor growth was measured with calipers starting on day 7 post-injection. Tumor volume was calculated using the formula: Tumor volume ($mm^3$) = length (mm) × width (mm) × height (mm) × 0.52. After 3 weeks, the mice were euthanized, and the xenografts were isolated. Tumor weights and volumes were recorded at the experimental endpoint.

## Clinical sample analysis

The formalin-fixed, paraffin-embedded pathological sections containing 15 paired tumor and paracancerous tissues from patients with colorectal cancer (CRC) were obtained from the Department of Oncology, the First Affiliated Hospital of Anhui Medical University. The Institutional Ethics Committees of the First Affiliated Hospital of Anhui Medical University approved the sample collection (AMUFAH-EC-PJ-2024-10-17). Patient information is provided in the corresponding source data.

The sections were deparaffinized at 65 °C for 1 h, followed by treatment with xylene and rehydration using graded ethanol. Antigen retrieval was conducted in a 95 °C water bath for 45–60 min in 0.01 M citrate buffer (pH 6.0) to remove aldehyde links formed during initial tissue fixation. Subsequently, immunofluorescence was performed. Images were acquired using a Nikon Eclipse Ti-E NIS-Elements AR software (version 4.30.1.10210). The intensity of LDHA and LDHB staining was quantified and scored as cancer versus normal tissue. The intensity of phosphorylated LDHA[T18] staining was assessed and graded by comparing mitotic cells relative to adjacent interphase cells. To minimize bias, data collection was conducted using a double-blind methodology.

## Culture of patient-derived intestinal organoids and infection with lentivirus

Human colorectal cancer organoids were purchased from AIMINGMED and plated as per the manufacturer's protocols. Following complete solidification of Matrigel (Corning, 356231), human colorectal cancer organoid complete medium (AIMINGMED, 10-100-018) was carefully added and refreshed every two days. The protocol for infecting human colorectal tumor organoids has been previously described (Bolhaqueiro et al, 2018). Briefly, lentivirus was concentrated by ultracentrifugation at $175,000 \times g$ and 4 °C for 90 min (Beckman, Optima MAX-XP). Human colorectal tumor organoids were digested with Gentle Cell Dissociation Reagent (Stemcell, 7174)

and infected with lentivirus in medium containing polybrene and ROCK inhibitor Y-27632 to enhance transduction efficiency and prevent anoikis. Infection was carried out in a cell incubator (37 °C, 5% $CO_2$) for 6 h. Subsequently, 50 μL of the organoid suspension in Matrigel was plated in a prewarmed 24-well plate, followed by the addition of organoid complete medium with Y-27632. For selection, the organoid medium was supplemented with 2 μg/mL puromycin 2–3 days post-infection.

## Statistical analysis

Data were from three or more independent experiments. Statistical analyses were carried out by GraphPad Prism v8.0.2 software (La Jolla, CA, USA). The data were presented as mean ± SD or violin plots. Differences between the two groups were analyzed by a two-tailed Student's *t*-test. Comparisons between more than two variables were analyzed by two-way ANOVA. Statistical significance is displayed as *$P < 0.05$, **$P < 0.01$, ***$P < 0.001$, ****$P < 0.0001$, and ns for not significant.

# Data availability

No large primary datasets have been generated and deposited for this study.

The source data of this paper are collected in the following database record: biostudies:S-SCDT-10_1038-S44319-025-00573-8.

# Peer review information

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

## Acknowledgements

This work was supported by the National Science Foundation of China (31970670 and 32170736 to J.G. and 92357301 to Z.Y.); The Strategic Priority Research Program of the Chinese Academy of Sciences, Grant XDB0940101 to Z.Y.; National Key R&D Program of China (2022YFA1303100 to X.Y.); the Fundamental Research Funds for the Central Universities (YD9100002028 to K.R.); Center for Advanced Interdisciplinary Science and Biomedicine of IHM of USTC (QYPY20220017 to Z.Y.); the "Laboratory for Synthetic Chemistry and Chemical Biology" under the Health@InnoHK Program by the Innovation and Technology Commission of Hong Kong (funding support to J.S.). All correspondence and requests for materials should be addressed to J.G. (jguo2013@ustc.edu.cn).

## Author contributions

**Mengting Liu**: Data curation; Formal analysis; Investigation; Methodology. **Aoxing Cheng**: Conceptualization; Data curation; Formal analysis; Validation; Investigation; Methodology. **Weiyi You**: Investigation; Methodology. **Chenxu Dai**: Data curation; Investigation. **Ting Wang**: Data curation; Investigation. **Ying Wu**: Investigation. **Jiaxin Wu**: Investigation; Methodology. **Fumei Zhong**: Investigation; Methodology. **Jue Shi**: Conceptualization. **Yingying Du**: Resources; Methodology. **Zhonghuai Hou**: Investigation; Methodology. **Ping Gao**: Conceptualization; Methodology. **Ke Ruan**: Investigation; Methodology. **Yi Yang**: Resources; Methodology. **Yuzheng Zhao**: Conceptualization; Resources; Supervision; Methodology. **Kaiguang Zhang**: Resources; Supervision; Project administration. **Zhenye Yang**: Conceptualization; Data curation; Supervision; Funding acquisition; Writing—original draft; Writing— review and editing. **Jing Guo**: Conceptualization; Supervision; Funding acquisition; Investigation; Writing—original draft.

Source data underlying figure panels in this paper may have individual authorship assigned. Where available, figure panel/source data authorship is listed in the following database record: biostudies:S-SCDT-10_1038-S44319-025-00573-8.

## Disclosure and competing interests statement

The authors declare no competing interests.

# Expanded View Figures

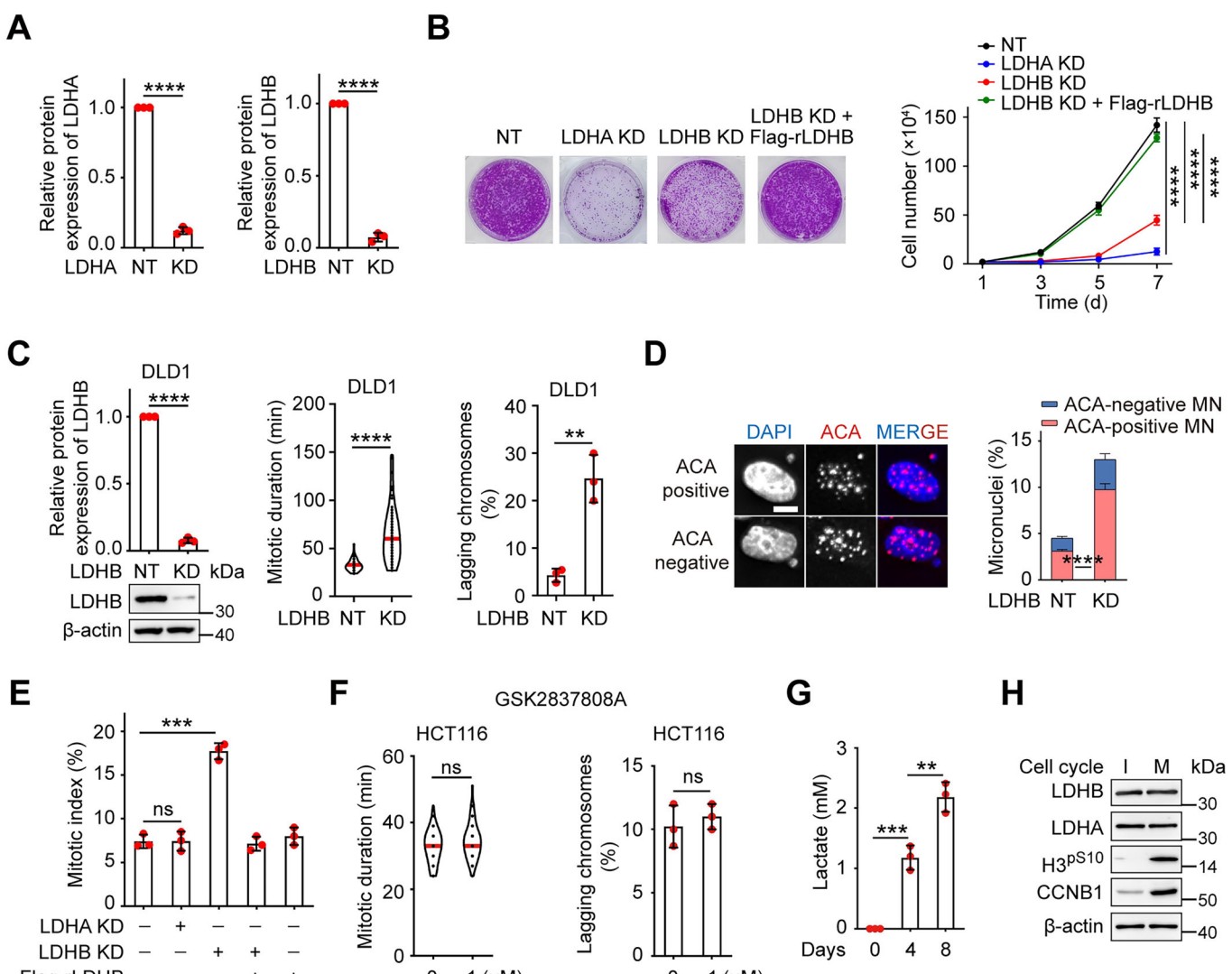

**Figure EV1. LDHB, not LDHA, is required for proper chromosome segregation.**

(A) The relative expression levels of LDHA and LDHB knockdown in Fig. 1A were analyzed ($n = 3$ biologically independent experiments; ****$P < 0.0001$). NT (non-targeting), KD (knockdown). (B) A colony formation assay was performed on the HeLa cells. After 10 days of culture, the cells were fixed and stained with crystal violet (left). A cell proliferation assay was conducted. Growth curves were plotted over a seven-day period ($n = 3$ biologically independent samples for each group; ****$P < 0.0001$) (right). (C) Cell lysates from DLD1 WT or LDHB KD cells were immunoblotted with the indicated antibodies, and the quantitative analysis is presented ($n = 3$ biologically independent experiments; ****$P < 0.0001$) (left). The mitotic duration was quantified ($n = 95, 95$ biologically independent cells; ****$P < 0.0001$), and quantification of lagging chromosomes is depicted ($n = 3$ biologically independent experiments; **$P = 0.0025$) (right). (D) HeLa WT and LDHB KD cells were stained with DAPI (blue) and anti-centromere antibodies (ACA) (red). Representative fluorescence images of centromere-positive and centromere-negative micronuclei are shown. Scale bar, 10 μm (left). The quantification of centromere-positive and centromere-negative micronuclei (MN) is presented ($n = 3$ biologically independent experiments; ****$P < 0.0001$) (right). (E) Cells were synchronized using a double thymidine block procedure and then fixed for DAPI staining 9 h after release. The mitotic index of each group was quantified ($n = 3$ biologically independent experiments; (ns) $P = 0.9679$, ***$P = 0.0001$). (F) HCT116 cells (H2B–GFP-labeled) treated with or without LDHA inhibitor GSK2837808A were monitored by time-lapse microscopy. The mitotic duration (min) starting from nuclear envelope breakdown to anaphase onset was quantified ($n = 71, 71$ biologically independent cells; (ns) $P = 0.0645$). Quantification of lagging chromosomes is shown ($n = 3$ biologically independent experiments; (ns) $P = 0.5313$). (G) Lactate levels in the culture medium of control organoids were measured at different culture time points using a lactate assay kit, and the medium was refreshed on day 4. ($n = 3$ biologically independent experiments; ***$P = 0.0005$, **$P = 0.0054$). (H) Western blot analysis was conducted to examine the expression of LDHA and LDHB in interphase and mitotic HeLa cells. Cyclin B1 (CCNB1) and H3$^{pS10}$ were utilized as mitotic markers. Immunoblots are representative of three independent experiments. Data Information: Data were shown as violin plots for (C) (middle), (F) (left), as mean ± SD for other data. Statistical significance was assessed by two-way ANOVA for (B); other data were determined by unpaired two-tailed Student's $t$-test.

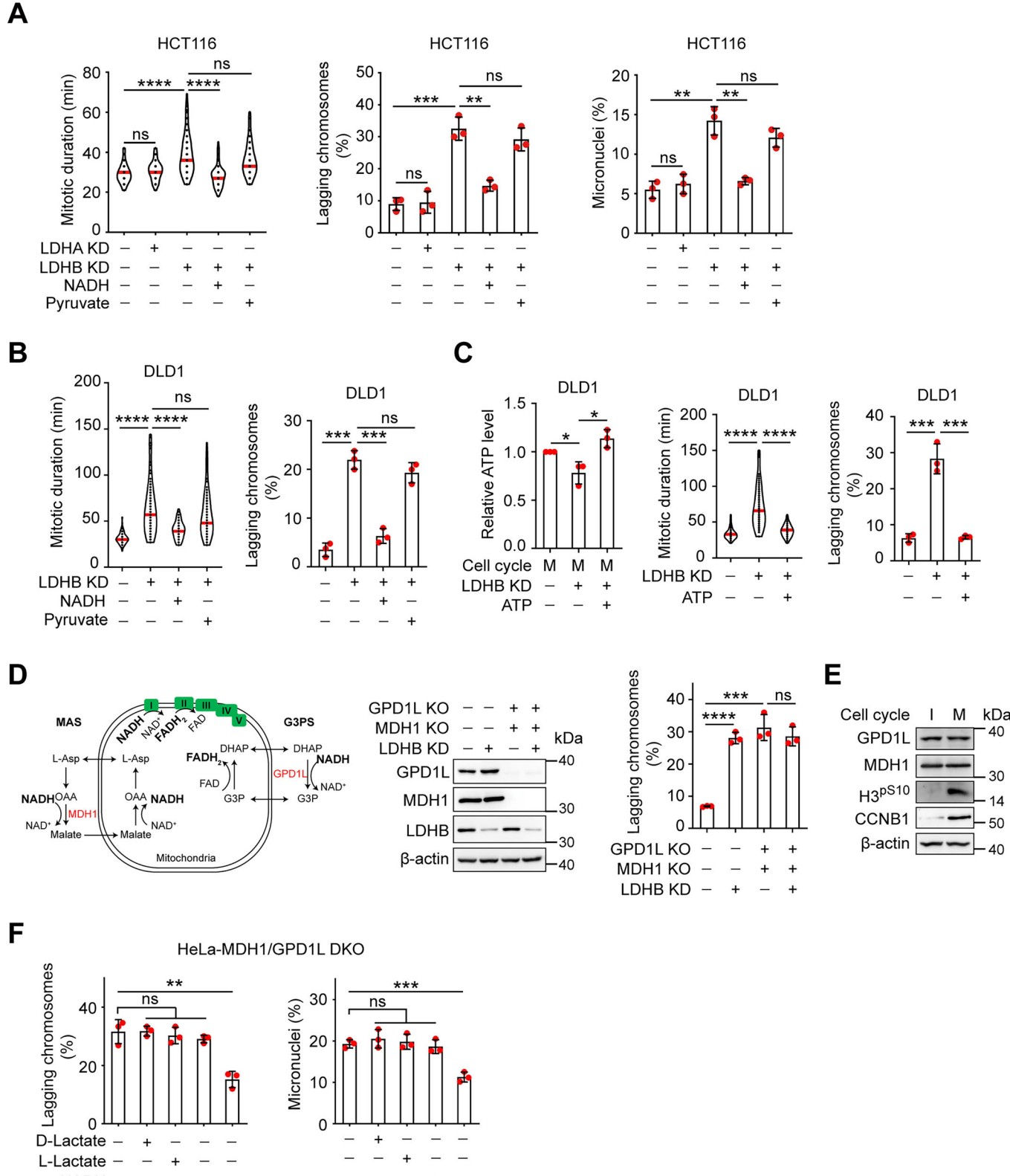

◀

**Figure EV2. The LDHB-mediated increase in mitotic NADH and ATP is essential for accurate chromosome segregation.**

(A) In HCT116-GFP-H2B cells, LDHA KD and LDHB KD cells treated with or without 1 mM NADH, 1 mM pyruvate were monitored using time-lapse microscopy. The mitotic duration was quantified ($n = 89, 89, 89, 89, 89$ biologically independent cells; (ns) $P = 0.0544$, ****$P < 0.0001$, (ns) $P = 0.0819$, from left to right) (left). The quantification of lagging chromosomes is presented ($n = 3$ biologically independent experiments; (ns) $P = 0.8262$, ***$P = 0.0006$, **$P = 0.0015$, (ns) $P = 0.3155$, from left to right) (middle). The quantification of micronuclei is shown ($n = 3$ biologically independent experiments; (ns) $P = 0.4795$, **$P = 0.0020$, **$P = 0.0020$, (ns) $P = 0.1592$, from left to right) (right). (B) In DLD1-GFP-H2B cells, LDHB KD cells were treated with 1 mM NADH or 1 mM pyruvate and monitored using time-lapse microscopy. The mitotic duration was quantified ($n = 116, 113, 100, 112$ biologically independent cells; ****$P < 0.0001$, (ns) $P = 0.0507$) (left). The quantification of lagging chromosomes is presented ($n = 3$ biologically independent experiments; ***$P = 0.0002$, ***$P = 0.0004$, (ns) $P = 0.1812$, from left to right) (right). (C) Relative changes of intracellular ATP levels in mitotic DLD1 cells upon LDHB KD or ATP supplementation were measured using an ATP assay kit ($n = 3$ biologically independent experiments; *$P = 0.0310$, *$P = 0.0142$) (left). Time-lapse microscopy was performed and mitotic duration was quantified ($n = 80, 80, 80$ biologically independent cells; ****$P < 0.0001$) (middle). The quantification of lagging chromosomes is presented ($n = 3$ biologically independent experiments; ***$P = 0.0009$, ***$P = 0.0008$, from left to right) (right). (D) A schematic illustration of two mitochondrial NADH shuttles, the malate-aspartate shuttle (MAS) and glycerol-3-phosphate shuttle (G3PS), is presented. The cytosolic enzymes malate dehydrogenase 1 (MDH1) and glycerol-3-phosphate dehydrogenase 1-like (GPD1L) are key components of MAS and G3PS, respectively (left). Extracts of HeLa WT, LDHB KD or MDH1 and GPD1L DKO cells were immunoblotted with the indicated antibodies (middle). Cells were filmed via time-lapse microscopy, and the quantification of lagging chromosomes is presented ($n = 3$ biologically independent experiments; ****$P < 0.0001$, ***$P = 0.0005$, (ns) $P = 0.4001$) (right). (E) Western blot analysis was conducted to examine the expression of MDH1 and GPD1L in interphase and mitotic HeLa cells. Cyclin B1 (CCNB1) and H3$^{pS10}$ were utilized as mitotic markers. (F) In Hela-GFP-H2B cells, MDH1 and GPD1L were knocked out and treated with 0.5 mM D-lactate, 0.5 mM L-lactate, 1 mM NADH, and 1 mM ATP. Cells were subsequently monitored using time-lapse microscopy. Quantification of lagging chromosomes and micronuclei is shown ($n = 3$ biologically independent experiments; (ns) $P = 0.9543$, (ns) $P = 0.6606$, (ns) $P = 0.3659$, **$P = 0.0046$, (ns) $P = 0.4387$, (ns) $P = 0.6970$, (ns) $P = 0.5850$, ***$P = 0.0009$, from left to right). Data Information: Data in **A** (left), **B** (left), and **C** (middle) are shown as violin plots; as mean ± SD for other data. Statistical significance was assessed by an unpaired two-tailed Student's *t*-test. Source data are available online for this figure.

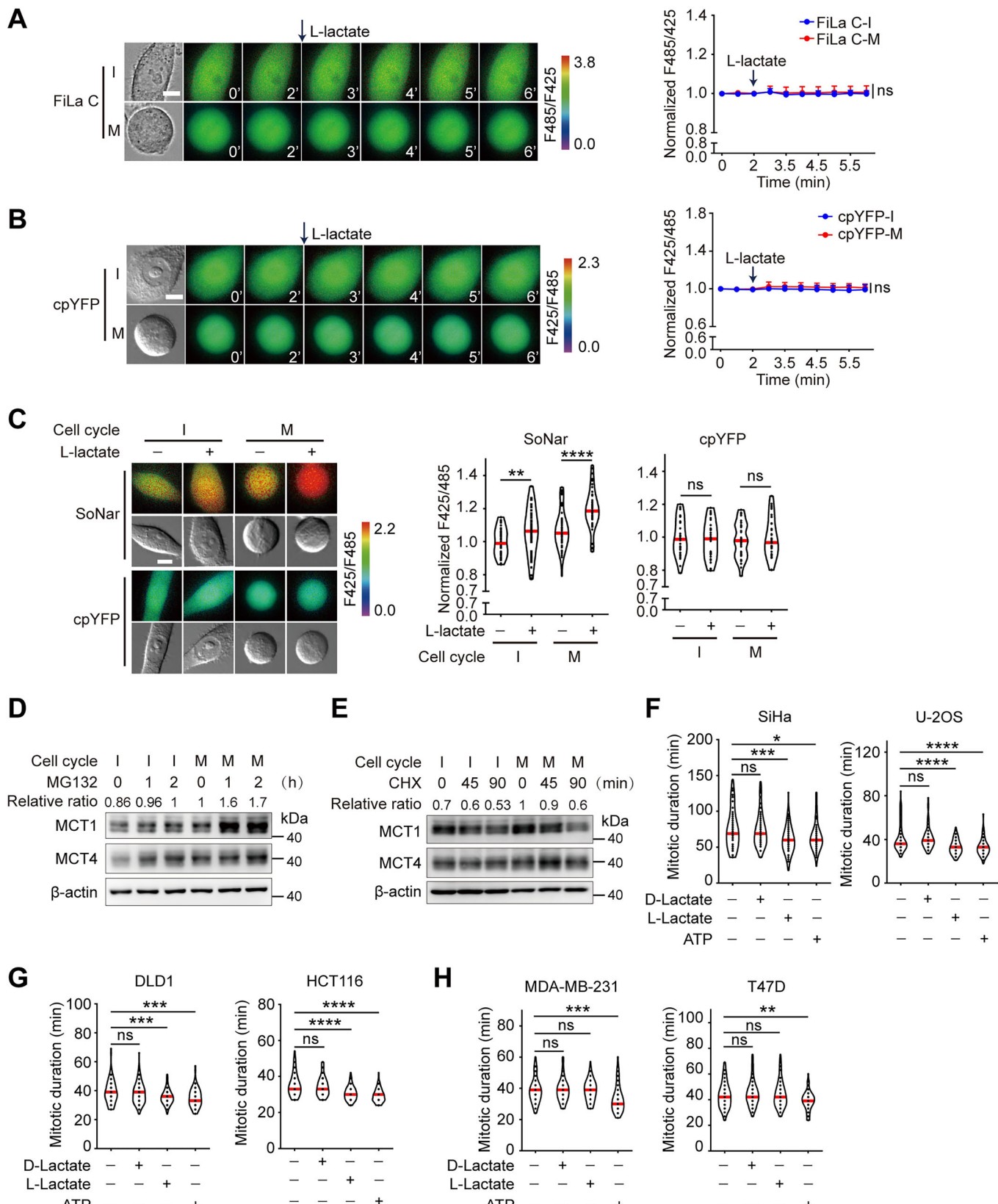

◄ **Figure EV3. MCT1-mediated lactate import increases during mitosis to ensure accurate chromosome segregation.**

(A) Time-lapse microscopy was performed to monitor the relative changes of intracellular lactate levels using FiLa C (control) in interphase and mitotic HeLa cells, both before and after treatment with 10 mM lactate. Images were pseudo-colored based on the F485/F425 ratio. Scale bar: 10 μm (left). Data were quantified over time ($n = 10$ biologically independent cells for interphase and $n = 11$ for mitosis; (ns) $P = 0.8637$) (right). (B) Time-lapse microscopy was conducted to measure the relative changes of intracellular NADH/NAD$^+$ ratios using cpYFP (control) in interphase and mitotic HeLa cells, both before and after treatment with 10 mM lactate. Images were pseudo-colored based on the F425/F485 ratio. Scale bar: 10 μm (left). Data were quantified over time ($n = 9$ biologically independent cells for interphase and $n = 8$ for mitosis; (ns) $P = 0.3218$) (right). (C) The plasmid of NADH/NAD$^+$ sensor SoNar and control sensor cpYFP were respectively transfected into HeLa cells. Cells were synchronized at interphase and mitosis using a double thymidine block procedure. Interphase and mitotic HeLa cells underwent live-cell imaging for 5–10 min in DMEM medium (Gibco, A1443001) supplemented with 10% FBS and 1% P/S. Subsequently, 10 mM lactate was added for an additional 5–10 min of imaging. Images were pseudo-colored based on the F425/F485 ratio. Scale bar: 10 μm (left). The relative intracellular NADH/NAD$^+$ ratios of each group are quantified ($n = 50, 50, 50, 50$ biologically independent cells for SoNar, $n = 34, 34, 34, 34$ biologically independent cells for cpYFP; **$P = 0.0083$, ****$P < 0.0001$, (ns) $P = 0.5419$, (ns) $P = 0.6498$, from left to right) (right). (D, E) Interphase and mitotic HeLa cells were obtained using a double thymidine block and shake-off. When cells progressed into interphase and mitosis, cells were treated with 10 μM MG132 or 100 μg/mL CHX for varying durations, followed by western blot analysis to examine the translational or protein stability regulation of MCT1. (F) SiHa/U-2OS (H2B–GFP-labeled) cells treated with 0.5 mM D-lactate, 0.5 mM L-lactate, 1 mM ATP were monitored using time-lapse microscopy. The mitotic duration was quantified ($n = 69, 69, 69, 69$ biologically independent cells for SiHa cells and $n = 74, 74, 85, 85$ for U-2OS cells; (ns) $P = 0.7957$, ***$P = 0.0005$, *$P = 0.0125$, (ns) $P = 0.8086$, ****$P < 0.0001$, from left to right). (G) DLD1/HCT116-GFP-H2B cells treated with 0.5 mM D-lactate, 0.5 mM L-lactate, and 1 mM ATP were monitored using time-lapse microscopy. The mitotic duration was quantified ($n = 90, 90, 90, 90$ biologically independent cells for DLD1 cells and $n = 89, 89, 89, 89$ for HCT116 cells; (ns) $P = 0.5033$, ***$P = 0.0004$, ***$P = 0.0001$, (ns) $P = 0.3690$, ****$P < 0.0001$, from left to right). (H) MDA-MB-231/T47D (H2B–GFP-labeled) cells treated with 0.5 mM D-lactate, 0.5 mM L-lactate, 1 mM ATP were monitored using time-lapse microscopy. The mitotic duration was quantified ($n = 67, 68, 68, 79$ biologically independent cells for MDA-MB-231 cells and $n = 71, 72, 72, 74$ for T47D cells; (ns) $P = 0.7768$, (ns) $P = 0.8532$, ***$P = 0.0002$, (ns) $P = 0.5361$, (ns) $P = 0.7743$, **$P = 0.0029$, from left to right). Data Information: Data in (C, F–H) are shown as violin plots; as mean ± SD for (A, B). Statistical significance was assessed by two-way ANOVA for (A, B); by unpaired two-tailed Student's $t$-test for (C, F–H).

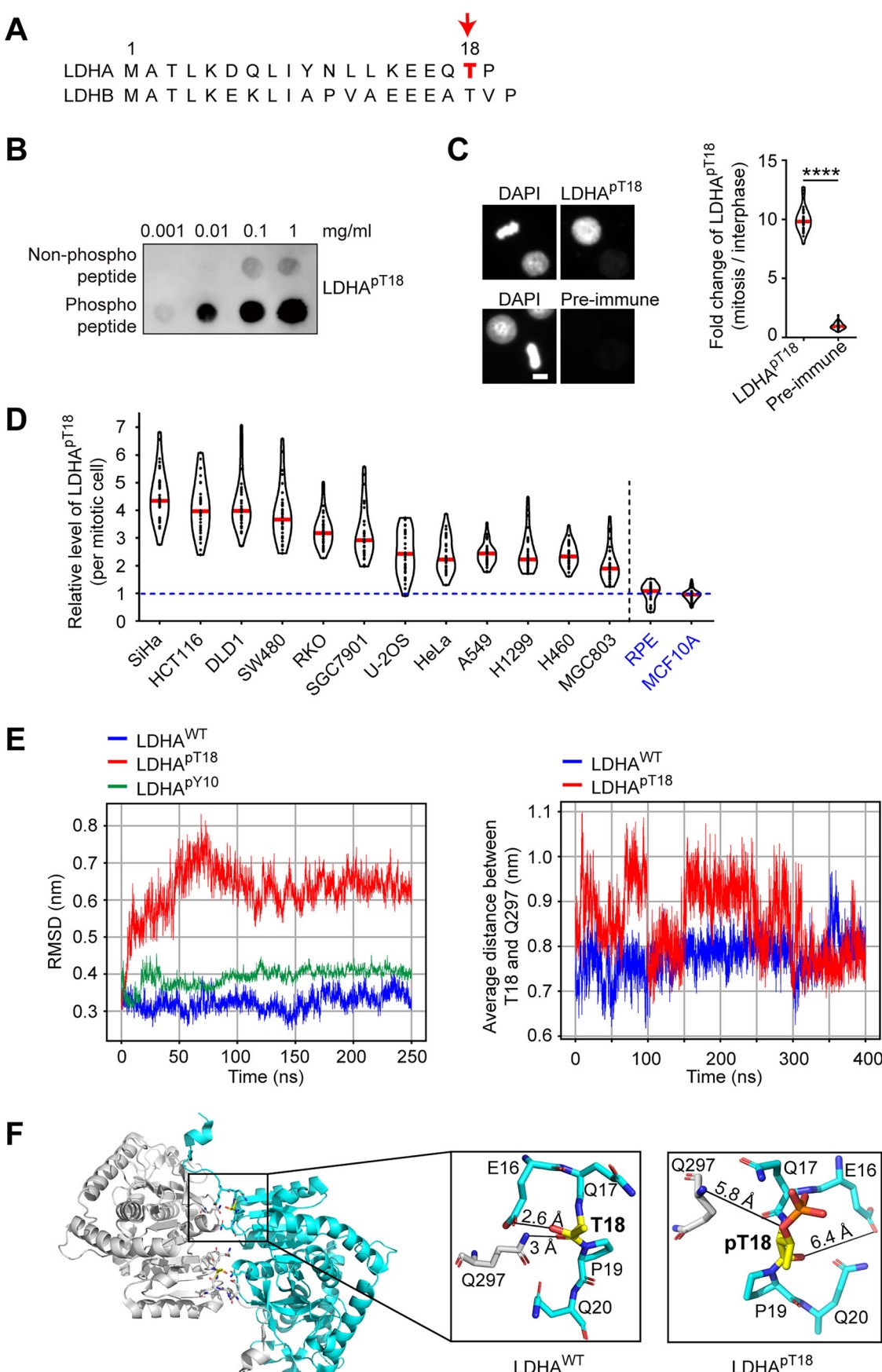

◀ **Figure EV4. Phosphorylation of LDHA at T18 alters LDH tetramer formation during mitosis.**

(A) An alignment of the 1–20 amino acid sequences of LDHA and LDHB proteins is presented. (B) The LDHA^T18-phosphorylated peptide and non-phosphorylated peptide were added to the NC membrane, followed by a dot blot assay with anti-LDHA^pT18 antibody. (C) Representative images of interphase and mitotic HeLa cells stained with anti-LDHA^pT18 antibody or pre-immune antibody. Scale bar, 10 μm (left). Relative fold change of LDHA^pT18 (mitosis/interphase) is shown ($n = 43$, 39 biologically independent cells; ****$P < 0.0001$) (right). (D) Quantified mitotic LDHA^pT18 level across various cancer cell lines and two non-cancer cells are shown ($n = 34$, 34, 34, 32, 34, 34, 38, 32, 35, 33, 33, 34, 31, 35 biologically independent cells, from left to right). (E) The root-mean-square deviation (RMSD) of LDHA^WT, LDHA^pT18 and LDHA^pY10 was analyzed over a 250 ns timeframe during molecular dynamics simulations, with their RMSD values represented in blue, red, and green, respectively (left). Additionally, the distance between residue T18 or pT18 of LDHA and residue Q297 of another LDHA monomer was measured over a 400 ns timeframe during molecular dynamics simulations (right). (F) Molecular dynamics simulations revealed the dynamic conformational changes of LDHA T18 or pT18 and Q297 of another LDHA monomer in 70 ns. The residues Q297 and T18 or pT18 of LDHA were highlighted in color. Data Information: Data in (C, D) are shown as violin plots. Statistical significance was assessed by an unpaired two-tailed Student's *t*-test. Source data are available online for this figure.

**A**

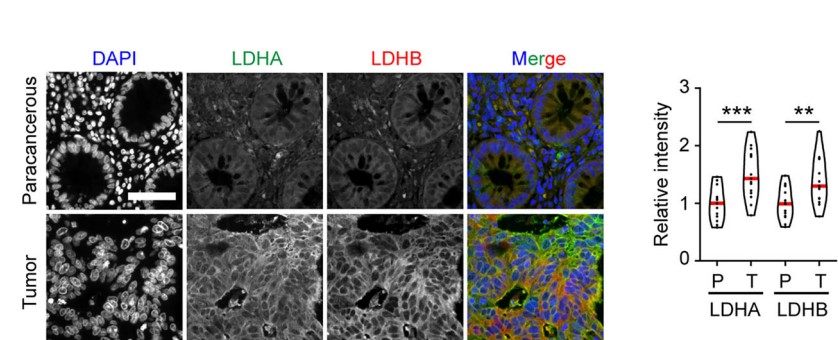

**Figure EV5. Phosphorylation of LDHA at T18 is essential for tumor progression.**

(**A**) Representative images of paracancerous and colorectal cancer (CRC) samples stained with DAPI (blue) and antibodies for LDHA (green) and LDHB (red). Scale bar, 50 μm (left). Quantification of relative expression levels of LDHA and LDHB in paired CRC samples is shown ($n$ = 15 biologically independent samples; ***$P$ = 0.0007, **$P$ = 0.0034), P: paracancerous, T: tumor (right). Data Information: Data in (**A**) is shown as violin plots. Statistical significance was assessed by an unpaired two-tailed Student's *t*-test.

