## [Peer Review File · EMBO Reports]

CDK1-Mediated Phosphorylation of LDHA Fuels Mitosis through LDHB-Dependent Lactate Oxidation

Mengting Liu, Aoxing Cheng, Weiyi You, Chenxu Dai, Ting Wang, Jiaxin Wu, Ying Wu, Fumei Zhong, Jue Shi, Yingying Du, Zhonghuai Hou, Ping Gao, Ke Ruan, Yi Yang, Yuzheng Zhao, Kaiguang Zhang, Zhenye Yang, and Jing Guo

Corresponding author(s): Zhenye Yang (zhenye@ustc.edu.cn), Yuzheng Zhao (yuzhengzhao@ecust.edu.cn), Jing Guo (jguo2013@ustc.edu.cn), Kaiguang Zhang (Zhangkaiguang@ustc.edu.cn)

Review Timeline:

Submission Date:	15th Sep 24
Editorial Decision:	14th Oct 24
Revision Received:	29th Jun 25
Editorial Decision:	5th Aug 25
Revision Received:	15th Aug 25
Accepted:	22nd Aug 25

Editor: Achim Breiling

Transaction Report:

Dear Dr. Yang,

Thank you for the submission of your manuscript to EMBO reports. I have now received the reports from the three referees that were asked to evaluate your study, which can be found at the end of this email.

As you will see, the referees find the study very interesting. Nevertheless, they have several comments, concerns, and suggestions, indicating that a major revision of the manuscript is necessary to allow publication of the study in EMBO reports. As the reports are below, and all the concerns need to be addressed, I will not detail them further here.

Given the constructive referee comments, I would like to invite you to revise your manuscript with the understanding that the concerns of the referees must be addressed in the revised manuscript and in a detailed point-by-point response. Acceptance of your manuscript will depend on a positive outcome of a second round of review. It is EMBO reports policy to allow a single round of revision only and acceptance of the manuscript will therefore depend on the completeness of your responses included in the next, final version of the manuscript.

- 1) a .docx formatted version of the final manuscript text (including legends for main figures, EV figures and tables), but without the figures included. Figure legends should be compiled at the end of the manuscript text.
- 2) individual production quality figure files as .eps, .tif, .jpg (one file per figure), of main figures and EV figures. Please upload these as separate, individual files upon re-submission.

- 4) a complete author checklist, which you can download from our author guidelines (<https://www.embopress.org/page/journal/14693178/authorguide>). Please insert page numbers in the checklist to indicate where the requested information can be found in the manuscript. The completed author checklist will also be part of the RPF.

- 5) that primary datasets produced in this study (e.g. RNA-seq, ChIP-seq, structural and array data) are deposited in an

appropriate public database. If no primary datasets have been deposited, please also state this in a dedicated section (e.g. 'No primary datasets have been generated and deposited'), see below.

The accession numbers and database should be listed in a formal "Data Availability" section that follows the model below. This is now mandatory (like the COI statement). Please note that the Data Availability Section is restricted to new primary data that are part of this study. This section is mandatory. As indicated above, if no primary datasets have been deposited, please state this in this section

Data availability

8) Regarding data quantification and statistics, please make sure that the number "n" for how many independent experiments were performed, their nature (biological versus technical replicates), the bars and error bars (e.g. SEM, SD) and the test used to calculate p-values is indicated in the respective figure legends (also for EV and Appendix figures). Please also check that all the p-values are explained in the legend, and that these fit to those shown in the figure. Please provide statistical testing where applicable. Please avoid the phrase 'independent experiment', but clearly state if these were biological or technical replicates. Please also indicate (e.g. with n.s.) if testing was performed, but the differences are not significant. In case n=2, please show the data as separate datapoints without error bars and statistics. See also: <http://www.embopress.org/page/journal/14693178/authorguide#statisticalanalysis>

9) Please add scale bars of similar style and thickness to microscopic images, using clearly visible black or white bars (depending on the background). Please place these in the lower right corner of the images themselves. Please do not write on or near the bars in the image but define the size in the respective figure legend.

10) Please also note our reference format:

12) We now use CRedit to specify the contributions of each author in the journal submission system. CRedit replaces the author contribution section. Please use the free text box to provide more detailed descriptions and do NOT provide your final manuscript text file with an author contributions section. See also our guide to authors: <https://www.embopress.org/page/journal/14693178/authorguide#authorshipguidelines>

13) All Materials and Methods need to be described in the main text using our 'Structured Methods' format, which is required for

all research articles. According to this format, the Methods section should include a Reagents and Tools Table (listing key reagents, experimental models, software, and relevant equipment and including their sources and relevant identifiers), uploaded as separate file, followed by a Methods section in which we encourage the authors to describe their methods using a step-by-step protocol format with bullet points, to facilitate the adoption of the methodologies across labs. More information on how to adhere to this format as well as downloadable templates (.doc) for the Reagents and Tools Table can be found in our author guidelines (section 'Structured Methods'):

14) Please order the manuscript sections like this, using these names:

Title page - Abstract - Keywords - Introduction - Results - Discussion - Methods - Data availability section - Acknowledgements (including funding information) - Disclosure and Competing Interests Statement - References - Figure legends - Expanded View Figure legends

15) Please make sure that all the funding information is also entered into the online submission system and that it is complete and similar to the one in the acknowledgement section of the manuscript text file.

I look forward to seeing a revised form of your manuscript when it is ready.

Yours sincerely,

Referee #1:

Liu et al. uncover a novel role for LDHB in preserving chromosomal integrity during mitosis, providing fresh insights into the metabolic regulation of the cell cycle. This study emphasizes the critical function of LDHB, often overshadowed by LDHA in cancer metabolism research. Notably, the authors demonstrate that CDK1 phosphorylates LDHA, inhibiting its tetramer formation, thereby shifting the balance toward LDHB activity. While the findings are valuable, key experiments and further clarifications are required to strengthen the conclusions and clearly differentiate this work from related studies.

Strengths:

1. **Novelty:** The study uncovers an underexplored function of LDHB in mitosis, emphasizing its distinction from LDHA, which is frequently discussed in the context of cancer metabolism. The focus on metabolic regulation of cell division is timely and significant.
2. **Potential Impact:** By identifying LDHB's role in mitotic progression, this study opens up new possibilities for therapeutic targeting in cancer, particularly in addressing the metabolic vulnerabilities of rapidly dividing cells. The link between LDHB activity, NADH production, and ATP generation provides a solid mechanistic foundation for understanding its role in cell cycle control.
3. **Robust Experimental Design:** The use of various models-HeLa and DLD1 cells, patient-derived organoids, and xenografts-supports the study's conclusions. The inclusion of live-cell imaging to capture mitotic defects adds clarity and depth to the analysis.
4. **Clinical Relevance:** The observation of increased LDHA phosphorylation in colorectal cancer samples connects the findings to human cancers, offering potential translational relevance.

Major requests:

1. **Comparisons to Chouchani's Work:** While the study does not directly overlap with Chouchani's recent Nature paper (PMID: 36921622), which demonstrates lactate's inhibition of SENP1, a more explicit differentiation would be beneficial. Specifically,

Chouchani's use of the LDHA-selective inhibitor GSK2837808A and conveniently ignored the role of LDHB. Giving a great opportunity for this paper to distinguish itself.

-Beyond the knockouts/knockdowns, please repeat your experiments using this inhibitor to help clarify whether the observed phenotypes are LDHB-specific, further distinguishing this work. You can then emphasize this point in the text.

2. Additional Controls: Also, from the Chouchani paper, the authors should include experiments with D-lactate and pyruvate supplementation, as controls, to contrast the L-lactate rescue. This distinction will strengthen the interpretation of the findings.

3. Rescue Experiments with Supplementation: The authors should perform rescue experiments using pyruvate, lactate, or NADH to determine whether the mitotic defects can be mitigated. This would allow the authors to distinguish lactate from NADH. If NADH can rescue in the absence of LDH but not lactate, this further distinguishes your paper from the Chouchani mechanism. I would then recommend changing the title of the paper to reflect this by including NADH in it.

4. Exploring Mitochondrial Involvement: There is growing evidence that LDHB and MCT1 may localize to mitochondria (PMID: 27618187, 38263463, 38924407). The authors should explore this possibility further, particularly given their own findings related to MCT1 expression. A closer investigation of mitochondrial lactate metabolism could yield important insights.

Specifically, I would recommend the following:

5. Glucose Consumption Measurements: The authors should measure changes in glucose uptake during mitosis. A decrease in glucose consumption could indicate NADH accumulation in the cytosol, while stable consumption may suggest mitochondrial NADH generation. These data would provide further context for understanding how NADH levels impact cell metabolism during mitosis.

6. MAS and G3PS Knockout Experiments: The authors should repeat these experiments in the presence of lactate to clarify the relative roles of cytosolic and mitochondrial NADH in mitotic progression. As the majority of Malate, Aspartate and glycerol-3-phosphate would be coming from glucose. This will help define the contributions of different cellular compartments to the observed phenotypes.

Minor Requests:

7. The selection criteria for tissue samples from TCGA for LDH expression analysis should be clearly explained. Why was the scope limited?

8. Please clarify whether lactate was present in the organoid media, and confirm whether LDHA experiments were conducted with organoids. If not, conducting these experiments would strengthen the overall conclusions, especially given the availability of these cells in the lab.

Rewrite of the Discussion Section: I strongly recommend a complete rewrite of the discussion to better highlight the novelty and significance of these findings. The current discussion downplays the study's impact, and a more comprehensive version should contextualize the results within the broader field, address remaining questions, and clearly distinguish the work from related studies, especially Chouchani's.

Conclusion: Liu et al. present important new findings on LDHB's role in mitosis and its distinction from LDHA. However, additional experiments and clarifications are necessary to fully establish the mechanisms involved and to distinguish this work from related studies. I thoroughly enjoyed reading this manuscript, and with these revisions, this study has the potential to make a significant contribution to the understanding of metabolic regulation during mitosis and its implications for cancer therapy.

Referee #2:

The authors demonstrate a new role for the regulation of the lactate and LDH complex in the control of mitotic progression. Although the findings are very interesting, most of the reported data do not show statistics or describe the statistical test used to compare the reported groups, making interpretation of the results difficult. For some of the experiments, there is no accessible information in the figure legend on how many cells were used for each quantification, and in some cases it appears that only one replicate was performed due to the lack of STDEV. The model chosen for micronuclei and lagging chromosomes is incorrect given that HeLa cells normally have high levels of these aberrations. In line with this, it is surprising that the authors observe only 5% of micronucleated cells in the basal condition. Furthermore, some of the statements are very general and would need to be rephrased, e.g. : Given that most nucleotide and lipid synthesis are accomplished in the S and G2 phases, the major metabolic demand during mitosis is ATP. I am not sure where the authors drew this conclusion.

Specific comments:

The TCGA analysis of LDH isoforms is at the transcriptional level. Can the author do this at the proteome level to see if the proteins themselves are upregulated?

How was the statistic performed? No information is given in the figure legend or methods.

HeLa micronuclei and anaphase bridges:

The results obtained are unexpected given that DMSO HeLa cells have only 5% mitotic defects (micronuclei). The experiment needs to be repeated using a karyotype stable cell line such as HTC116, which is typically used for mitotic studies.

The mitotic index of each condition should be added to confirm the mitotic defect. In addition, to determine whether micronuclei originate from chromosome lag in mitosis rather than DNA breakpoint in interphase, the authors should perform centromere staining and quantify the percentage of centromere-containing micronuclei and lagging chromosomes.

Figure 1B lacks controls: shNT and shNT+flag rLDHB HeLa cells should be added to the comparison.

NADH supplementation:

How is NADH transported inside cells? As far as I know, there is no membrane transporter for NADH, nor should it diffuse. Can the authors check that NADH (and pyruvate) are being transported correctly?

Figure 3C, are there no biological replicates? the STDEV is missing and nothing is specified in the figure legend

It is known that excessive intracellular lactate reduces glycolytic flux 17, therefore the high level of lactate in the mitotic cells might be likely from extracellular lactate. How does this make sense? Lactate imported from the extracellular environment will also reduce the glycolytic flux, so there is essentially no reason for cells to import it unless they are going to use it as a carbon source, otherwise it will block glycolysis.

Without AZD3965, addition of exogenous lactate led to a twofold increase of cytosolic lactate level during mitosis, which accelerated mitotic progression. I am not sure how the authors can talk about cytosolic lactate levels in mitosis, as there is no nucleus in mitosis.

How generalizable is the system?

What happens in cells that do not express LDHB, such as T47D cells?

What happens in cells that express low levels of MCT1, such as MDAMB231?

(Other cell line expressing low/high levels of either LDHs or MCTs can be found in the CCLE proteomics database)

It is not clear how the authors monitor lactate-mediated mitotic effects: how long after lactate treatment are cells followed in live cell imaging? How many cells per biological replicate? How are cells selected? all at mitotic entry? why did the author not arrest cells in G2 to facilitate analysis?

Is there any evidence that MCT1 transporters are regulated during mitosis? Are they cell cycle dependent genes? Can they be found in the 10.7554/eLife.01630 dataset? how do the authors explain that the expression of MCT1 increased in the 1 hour duration of mitosis?

In the images shown in Figure 4, LDHA pT18 appears to localise to the mitotic spindle and centrosomes, supporting a role in mitosis beyond the indirect role attributed to it by the authors.

Does LDHAT18A cause lagging chromosomes due to mislocalisation or disruption of its role in the mitotic spindle? Does the mitotic spindle show defects in cells expressing LDHAT18A?

CDK1-mediated phosphorylation of LDHA is required for mitotic progression and tumor growth -> this cannot be correct because LDHA repression does not have any mitotic defects, according to the authors

5C, colours should be shown in individual images. How many cells per paired sample were quantified? This is not specified, as is generally the case with many figures. The statistical test is also not mentioned.

5B: How many mitotic cells have been analysed per replicate? Where are the source data?

A recent publication in Nature showed that lactate controls the APC: the authors barely mention this in the discussion. How does their work relate to this publication? Are these two unrelated mechanisms by which lactate controls mitosis?

Does inhibiting the ETC have an effect on mitotic progression? Would it be different to inhibit the ETC at different levels? Does this regulation of lactate in mitosis change depending on the glycolytic state of the cells? To test whether ATP is the molecule required for mitosis progression and its dependence on NADH, the authors should treat cells with either oligomycin or BAM15. The latter is a mitochondrial uncoupler that increases electron transfer but slows down ATP synthesis.

ATP is above 1 mM in the cells. This is considered excessive for cellular requirements. What is the metabolic demand for ATP in mitotic cells that would justify a requirement for this shift in LDH complexes?

In general, I would recommend looking out for oversimplified conclusions, lack of statistical detail, lack of source data, lack of important experimental details such as number of biological replicates or cells analysed.

Referee #3:

In their paper "CDK1 reverses the catalytic direction of lactate dehydrogenase to fuel mitosis in cancers", the authors examine mitotic cell metabolism and especially how lactate metabolism changes in mitosis. In short, they identify LDHB critical for normal mitosis and this seems to be mediated by LDHB-mediated lactate oxidation that helps sustain cellular NADH levels. The authors identify CDK1 phosphorylation on LDHA in mitosis that is involved in the increased LDHB activity in mitosis. They also claim that

these effects result in increased ATP production to support cell division.

Overall, the authors do provide a clear and interesting advancement regarding our understanding of mitotic metabolism. Little is known about the regulation of lactate metabolism in mitosis and the results presented here will be of interest to the field. However, the manuscript has some major issues, as some of the conclusions are not justified by the data, some key controls are missing, and the results are not discussed in relation to existing literature. Correcting these issues is essential before this work can be published. However, addressing these issues should be relatively easy and doing so will increase the impact of the work, while also making the manuscript more transparent. Please see below for details.

Major concerns:

Conclusions about mitochondrial metabolism are not justified by the data

Across the manuscript (title, abstract, intro, results, several sections in the discussion, and Fig 5D) the authors imply that the metabolic changes they identify in mitosis will result in increased ATP synthesis that supports mitosis. Not only is this conclusion not justified by the data, but it's also questionable considering existing literature. The authors show no data on mitochondrial metabolic rates, not to mention ATP synthesis rates in mitochondria or glycolysis. The fact that ATP levels decrease in mitosis does not suffice to conclude about the metabolic fluxes (synthesis rates). The reported decrease in ATP levels (~30% decrease) is unlikely to influence any mitosis driving protein activity, as cellular ATP levels are over an order of magnitude higher than the K_m values of motor proteins etc. The fact that ATP supplementation rescues some of the mitotic defects observed by the authors is curious, but this does not prove that the changes in lactate metabolism drive ATP synthesis.

To address this, I think the authors have two options. They could carry out extensive metabolite tracing experiments and oligomycin-responsive oxygen consumption experiments to test how ATP synthesis rates change in mitosis. The authors should also be able to rescue the mitotic defects caused by LDHB KD by driving more NADH into mitochondria (e.g. via MDH1 and GPD1L overexpression). However, considering that the manuscript's key discoveries (CDK1 controlling lactate metabolism) are not reliant on these conclusions about ATP synthesis and mitochondrial metabolism, the authors could just remove all these claims from the manuscript. The work would still be sufficiently interesting for publication in EMBO reports, and the work might be an even better fit for the journal's requirement that "conclusions constitute a single key message worthy of publication in EMBO reports".

Key experimental controls are missing and reproducibility is unclear:

- The authors need to validate the specificity of the antibody they've generated to detect phosphorylated LDHA, and these validations need to be shown in the supplements. The authors need to compare control and LDHA KD cells. I would expect to see IF images of these cells labelled with the antibody along with quantifications of the signal intensities. I would also expect to see a full western blot membrane following labeling with this antibody. In addition, there are currently no method details about the antibody generation or validation, and this needs to be corrected.
- Based on the methods section, most experiments are done by comparing G1/S synchronized cells to M-phase synchronized cells. I'm glad to see that the authors used mitotic shake-off for the enrichment of M-phase cells, as this is likely to yield more 'normal' mitotic cells than mitotic arrests. However, key experiments should include a comparison to unsynchronized cells and/or an alternative interphase enrichment method (e.g. leftovers from the mitotic shake-off). Otherwise, the conclusions could be due to changes specifically in G1/S transition rather than M-phase. In addition, the authors need to clarify when were cells synchronized to mitosis using mitotic arrest (as done in Fig 4E), and how exactly this was done.
- The consistency of LDHA and LDHB knockdowns is unclear. In Fig S1B, the LDHB protein band disappears with the KD, but then reappears partly when Flag-rLDHB is introduced (i.e. the smaller band on the blot is probably endogenous LDHB). This suggests that the KD is not always complete nor consistent. The authors should quantify the KD efficiency across experiments and cell lines and show this in the supplements.
- Fig 1F is critical for this paper and its key conclusions. While the blot shown is convincing, the reproducibility of this finding is unclear. How many times has this experiment been replicated? Please show quantifications across experiments and, preferably, in different cell lines too. Similarly, additional data from a different cell line would strengthen the Fig 1E, which is also critical to this manuscript.
- Optimally, the reverse activity of the LDH isoenzyme would be verified with metabolite tracing experiments. This would significantly increase the credibility of this work. However, such assays may not be easily available to the authors, and if this validation cannot be done the lack of it should be acknowledged in the text.
- Fig 1 is missing some controls. What is the typical mitotic behavior of WT cells in Fig 1B? How efficient is the LDHB knockdown in Fig 1C when working with organoids?
- The MDH1 and GPD1L are knocked out, but the western blot in 2F shows weak bands of the right size. This should not happen if the KO is successful. There are no method details on how the KO was carried out. The KOs need to be validated and described properly.
- The results of fluorescence-based metabolite sensors need clarification. In Fig 3C,D, how many cells were examined and in how many experiments? Are the sensor responses linear with the metabolite change? If not, the comparison between interphase and mitotic cells using the sensors (Fig 3C,D) seem questionable.
- In Fig 3G, the authors should consider also showing how intracellular lactate levels change with the MCT knockdowns.
- The *in silico* analyses in Fig 4J,K require controls that allow the evaluation of effect specificity and magnitude. Would phosphorylation on any phospho-site result in similar effects?

Existing literature is ignored, which influences the interpretation of the data

There are several publications from recent years that are not acknowledged by the authors, even though these publications could help interpret the authors' work. This limits the impact of the work and makes some aspects of the writing seem controversial.

- Mitochondrial ETC activity seems to be increased in mitosis, but ATP synthesis may decrease because mitochondrial ATP synthase is partially inhibited (1)
- Glycolytic fluxes may be directed away from pyruvate in mitosis (2)
- ATP level decrease in mitosis is well documented (1, 3, 4)
- The authors also suggest that biosynthesis rates in mitosis are low and energy is used for cell division. While there's no clear evidence that cell division requires a lot of energy, there are many papers showing that biosynthesis remains nearly at interphase levels during normal mitosis (when cells are not arrested in mitosis) (5-7)

Is the LDHA T18 phosphorylation typical in normal cells?

In Fig 5C, there is little LDHA T18 phosphorylation in the non-tumor tissue. This should be acknowledged and discussed, as it seems like this phosphorylation may not be a feature of normal, non-cancerous cells. Optimally, the authors would test the LDHA T18 phosphorylation levels in several normal and cancerous models (either in vivo or just cell lines), but at minimum this needs to be acknowledged in the text.

Minor points:

- I recommend noting in the results section how the cells were synchronized, as this is important for the experimental outcome.
- In figure legends, the authors use the expression "biologically independent experiments/samples". This should be clarified somewhere. Were these independent cultures grown and measured at the same time? Or were these separate experiments from different days? And is there a difference between experiments and samples when detailing these?
- The manuscript could be shortened by moving Figures 1A, 2F, 4A, 4J, 4K to the supplement. Also the discussion could be cut shorter by removing repetitive detailing of the findings and by cutting away parts of the second to last paragraph.
- How did the authors select the cancer types to evaluate in Fig 1A? Are there potential sampling biases?
- When describing Fig 1E results in the text, authors claim 2-fold increase in lactate to pyruvate conversion rates. This is an overstatement based on the data figure.
- The results could be informative of why cancer cells undergo the Warburg effect. This seems completely ignored in the discussion section.

References:

1. J. H. Kang, et al., Monitoring and modeling of lymphocytic leukemia cell bioenergetics reveals decreased ATP synthesis during cell division. *Nat Commun* 11, 4983 (2020).
2. X. Ma, et al., Polo-like kinase 1 coordinates biosynthesis during cell cycle progression by directly activating pentose phosphate pathway. *Nat Commun* 8 (2017).
3. K. Maeshima, et al., A transient rise in free Mg(2+) ions released from ATP-Mg hydrolysis contributes to mitotic chromosome condensation. *Current Biology* 28, 444-451 (2018).
4. E. Doménech, et al., AMPK and PFKFB3 mediate glycolysis and survival in response to mitophagy during mitotic arrest. *Nat Cell Biol* 17, 1304-1316 (2015).
5. T. P. Miettinen, J. H. Kang, L. F. Yang, S. R. Manalis, Mammalian cell growth dynamics in mitosis. *Elife* 8 (2019).
6. V. Stonyte, E. Boye, B. Grallert, Regulation of global translation during the cell cycle. *J Cell Sci* 131 (2018).
7. M. Shuda, et al., CDK1 substitutes for mTOR kinase to activate mitotic cap-dependent protein translation. *Proc Natl Acad Sci U S A* 112, 5875-5882 (2015).

We sincerely thank the Editor and Reviewers for their insightful critiques and constructive suggestions. These comments guided essential additional experiments and analyses that have strengthened our data, reinforced key conclusions, and elevated the overall impact of the manuscript. Below, we provide detailed point-by-point responses (**in blue text**) to each concern raised, with corresponding revisions in the manuscript and supplementary materials.

Point-by-point response to referee's comments

Reviewer #1 (Remarks to the Author):

Major comments:

1. Comparisons to Chouchani's Work: While the study does not directly overlap with Chouchani's recent Nature paper (PMID: 36921622), which demonstrates lactate's inhibition of SENP1, a more explicit differentiation would be beneficial. Specifically, Chouchani's use of the LDHA-selective inhibitor GSK2837808A and conveniently ignored the role of LDHB. Giving a great opportunity for this paper to distinguish itself.

-Beyond the knockouts/knockdowns, please repeat your experiments using this inhibitor to help clarify whether the observed phenotypes are LDHB-specific, further distinguishing this work. You can then emphasize this point in the text.

Response: We sincerely thank the reviewer for their constructive suggestions. In the revised manuscript, we have incorporated the use of the LDHA inhibitor GSK2837808A in time-lapse assays to assess mitotic progression following LDHA inhibition. HeLa and HCT116 cells were treated with GSK2837808A, and the results showed that LDHA inhibition did not lead to mitotic arrest or the appearance of lagging chromosomes (**Fig. 1B and EV1F**). These findings suggest that LDHA is not essential for mitosis in these two cell lines. Additionally, in line with the reviewer's recommendation, we have now included a comparison of our findings with those reported by Chouchani and colleagues in the Discussion section.

2. Additional Controls: Also, from the Chouchani paper, the authors should include experiments with D-lactate and pyruvate supplementation, as controls, to contrast the L-lactate rescue. This distinction will strengthen the interpretation of the findings.

Response: In accordance with the reviewer’s suggestion, we included D-lactate and pyruvate as controls in the mitotic progression experiment (Fig. 3F). Our results demonstrate that only NADH and L-lactate were able to accelerate mitotic progression, whereas D-lactate and pyruvate did not promote mitotic progression. These findings highlight a specific role for L-lactate in supporting mitosis and suggest a distinct metabolic function of lactate during this phase of the cell cycle. This mechanism appears to diverge from that described by Chouchani et al., pointing to a novel aspect of lactate metabolism in the regulation of mitosis.

3. Rescue Experiments with Supplementation: The authors should perform rescue experiments using pyruvate, lactate, or NADH to determine whether the mitotic defects can be mitigated. This would allow the authors to distinguish lactate from NADH. If NADH can rescue in the absence of LDH but not lactate, this further distinguishes your paper from the Chouchani mechanism. I would then recommend changing the title of the paper to reflect this by including NADH in it.

Response: In accordance with the reviewer’s suggestion, we included NADH, L-lactate, D-lactate, and pyruvate as controls in the rescue experiment (Fig. 2B). Our results demonstrate that only NADH was capable of rescuing the mitotic delay induced by LDHB depletion, whereas L-lactate, D-lactate, and pyruvate failed to restore normal mitotic progression. These findings underscore a specific and critical role for NADH in supporting mitosis under LDHB-deficient conditions, and reveal a unique aspect of lactate metabolism during mitosis that diverges from the mechanisms reported by Chouchani et al. In response to the reviewer’s recommendation, we have also revised the title, however due to character limitation, “NADH” could not be incorporated.

4. Exploring Mitochondrial Involvement: There is growing evidence that LDHB and MCT1 may localize to mitochondria (PMID: 27618187, 38263463, 38924407). The authors should explore this possibility further, particularly given their own findings related to MCT1 expression. A closer investigation of mitochondrial lactate metabolism could yield important insights.

Response: We agree with the reviewer that mitochondrial-localized LDHB and MCT1 likely contribute to lactate metabolism. However, as suggested by Reviewer 3, we have chosen to focus on the regulation of cytosolic LDH in order to streamline the narrative and avoid introducing unnecessary complexity. Accordingly, we have tempered our claims regarding mitochondrial involvement in this study. We acknowledge that fully elucidating the functional roles of mitochondrial LDHB and MCT1 will require extensive further investigation. As such, we have reduced the emphasis on mitochondrial mechanisms in the Results section. Nonetheless, we have addressed these considerations and highlighted the key outstanding questions in the Discussion section.

Specifically, I would recommend the following:

5. Glucose Consumption Measurements: The authors should measure changes in glucose uptake during mitosis. A decrease in glucose consumption could indicate NADH accumulation in the cytosol, while stable consumption may suggest mitochondrial NADH generation. These data would provide further context for understanding how NADH levels impact cell metabolism during mitosis.

Response: We measured glucose uptake in both interphase and mitotic cells and found that glucose consumption slightly increases during mitosis. This observation suggests that NADH generation and consumption are likely maintained through mitochondrial metabolism during mitosis.

6. MAS and G3PS Knockout Experiments: The authors should repeat these experiments in the presence of lactate to clarify the relative roles of cytosolic and mitochondrial NADH in mitotic progression. As the majority of Malate, Aspartate and glycerol-3-phosphate would be coming from glucose. This will help define the contributions of different cellular compartments to the observed phenotypes.

Response: We performed metabolite supplementation assays in MAS and G3PS dual knockout cells. Notably, neither lactate nor NADH was able to rescue the mitotic defects caused by

MDH1/GPD1L depletion, whereas ATP supplementation effectively restored mitotic progression (**Fig. EV2F**). These findings indicate that mitochondrial electron transport chain (ETC)-dependent ATP synthesis is the essential downstream mechanism of lactate/NADH metabolism required to support mitosis.

Minor Requests:

7. The selection criteria for tissue samples from TCGA for LDH expression analysis should be clearly explained. Why was the scope limited?

Response: We initially selected tissue samples and cell lines from the TCGA dataset and several published studies based on high expression levels of both LDHA and LDHB (PMID: 38291366; PMID: 38709280). These references have been cited in the Introduction sections of the revised manuscript. However, in accordance with Reviewer #3’s suggestion to streamline the manuscript, we have removed these data from the current version.

8. Please clarify whether lactate was present in the organoid media, and confirm whether LDHA experiments were conducted with organoids. If not, conducting these experiments would strengthen the overall conclusions, especially given the availability of these cells in the lab.

Response: We measured lactate concentrations in the organoid culture media (**Fig. EV1G**) and confirmed the presence of lactate. Additionally, we performed LDHA knockdown assays in organoids and observed that LDHA depletion inhibited organoid growth (**Fig. 1C**), consistent with the phenotype observed in cultured cell lines (**Fig. EV1B**).

9. Rewrite of the Discussion Section: I strongly recommend a complete rewrite of the discussion to better highlight the novelty and significance of these findings. The current discussion downplays the study's impact, and a more comprehensive version should contextualize the results within the broader field, address remaining questions, and clearly distinguish the work from related studies, especially Chouchani's.

Response: We have revised the Discussion section to emphasize both the novelty and significance of our findings. In addition, we have incorporated a detailed comparison with the study by Chouchani et al., highlighting the key mechanistic differences and contextual relevance of our work.

Conclusion: Liu et al. present important new findings on LDHB's role in mitosis and its distinction from LDHA. However, additional experiments and clarifications are necessary to fully establish the mechanisms involved and to distinguish this work from related studies. I thoroughly enjoyed reading this manuscript, and with these revisions, this study has the potential to make a significant contribution to the understanding of metabolic regulation during mitosis and its implications for cancer therapy.

Reviewer #2 (Remarks to the Author):

The authors demonstrate a new role for the regulation of the lactate and LDH complex in the control of mitotic progression. Although the findings are very interesting, most of the reported data do not show statistics or describe the statistical test used to compare the reported groups, making interpretation of the results difficult. For some of the experiments, there is no accessible information in the figure legend on how many cells were used for each quantification, and in some cases it appears that only one replicate was performed due to the lack of STDEV. The model chosen for micronuclei and lagging chromosomes is incorrect given that HeLa cells normally have high levels of these aberrations. In line with this, it is surprising that the authors observe only 5% of micronucleated cells in the basal condition. Furthermore, some of the statements are very general and would need to be rephrased, e.g. : Given that most nucleotide and lipid synthesis are accomplished in the S and G2 phases, the major metabolic demand during mitosis is ATP. I am not sure where the authors drew this conclusion.

Response: We are grateful for the reviewers' constructive comments. In the revised manuscript, we have carefully addressed all the raised concerns, which has significantly improved the clarity and overall quality of our study.

Specific comments:

1. The TCGA analysis of LDH isoforms is at the transcriptional level. Can the author do this at the proteome level to see if the proteins themselves are upregulated?

Response: We thank the reviewer for raising this important point regarding protein-level expression. To address this, we thoroughly examined protein expression data from multiple sources, including TCGA (linked protein data), the Human Protein Atlas, the CPTAC database, and relevant literature (PMID: 38291366; PMID: 38709280). Our analysis revealed that LDHA and LDHB protein expression in human patient samples and cell lines is highly variable and context-dependent. However, a significant limitation we encountered is that the available data—derived primarily from mass spectrometry—generally cannot reliably distinguish between the LDHA and LDHB isoforms (The amino acid sequence similarity between LDHA and LDHB is 75%). While we confirmed high expression levels for both isoforms collectively in the cells we tested, the lack of isoform-specific resolution at the protein level prevented us from conducting a robust comparative proteomic analysis analogous to our transcriptional TCGA analysis. Consequently, and in response to the collective feedback from all three reviewers on this issue, we have removed the transcriptional TCGA analysis of LDH isoforms from the revised manuscript.

2. How was the statistic performed? No information is given in the figure legend or methods.

Response: We have improved the statistical analysis by providing additional details and incorporating more experimental data. Corresponding figure legends and the Methods section have also been revised accordingly to ensure clarity and completeness.

3. HeLa micronuclei and anaphase bridges:

The results obtained are unexpected given that DMSO HeLa cells have only 5% mitotic defects (micronuclei). The experiment needs to be repeated using a karyotype stable cell line such as HCT116, which is typically used for mitotic studies.

Response: We thank the reviewer for highlighting this important point regarding baseline mitotic defects in HeLa cells and for suggesting validation in karyotypically stable models. While published studies report variable baseline micronuclei frequencies in HeLa (5–15%; PMID: 38749421, 31320611, 35393399, 34010649, 39737931), we acknowledge that this heterogeneity could reflect methodological differences in cell culture, imaging parameters (e.g., z-axis resolution affecting micronuclei detection), or intrinsic genomic instability.

To directly address this concern and follow the reviewer’s recommendation, we repeated all key experiments in two karyotypically stable cell lines: HCT116 and DLD1. The results from these orthogonal validations—now included as **Fig. EV1C, EV1E, Fig. EV2A-C and Fig. EV3G** in the revised manuscript—fully corroborate our original findings in HeLa cells. This consistency across distinct cellular models strengthens the biological relevance of our conclusions.

4. The mitotic index of each condition should be added to confirm the mitotic defect.

Response: Information on the mitotic index has been included in the revised manuscript and is presented in **Fig. EV1E**. Consistent with the imaging data, only LDHB knockdown resulted in a significant increase in mitotic index.

5. In addition, to determine whether micronuclei originate from chromosome lag in mitosis rather than DNA breakpoint in interphase, the authors should perform centromere staining and quantify the percentage of centromere-containing micronuclei and lagging chromosomes.

Response: We thank the reviewer for this insightful suggestion to determine whether micronuclei originate from mitotic chromosome lagging versus interphase DNA breakage. To directly address this mechanistic question, we performed centromere immunostaining using anti-centromere antibodies (ACA). As shown in **Fig. EV1D**, we quantified the proportion of micronuclei containing centromeres (ACA-positive). The high frequency of ACA-positive micronuclei provides direct evidence that most micronuclei in our system derive from chromosome segregation errors during mitosis.

6. Figure 1B lacks controls: shNT and shNT+flag rLDHB HeLa cells should be added to the comparison.

Response: The controls has been included in the revised manuscript (**Fig. 1A**)

7. NADH supplementation:

How is NADH transported inside cells? As far as I know, there is no membrane transporter for NADH, nor should it diffuse.

Can the authors check that NADH (and pyruvate) are being transported correctly?

Response: We thank the reviewer for raising this important mechanistic question regarding NADH transport. While no dedicated NADH transporter has been identified, our data demonstrate functional intracellular delivery: Direct measurement shows significantly increased intracellular NADH levels following extracellular NADH supplementation (**Fig. 2F**, new panel added). Which is consistent with prior evidence in Cell Metabolism (PMID: 25955212). In addition to a direct transporter, these functional NADH uptake may occur via: Endocytosis, Redox shuttles (e.g., membrane-associated transhydrogenases) or Indirect electron/proton transfer. Pyruvate is rapidly transported into cells via monocarboxylate transporters (MCTs) after addition into the culture medium, as extensively documented (e.g., Elife. 2020. PMID: 32142409).

8. Figure 3C, are there no biological replicates? the STDEV is missing and nothing is specified in the figure legend

Response: We have included additional biological replicates in the assay, and the corresponding statistical analyses have been incorporated into the revised manuscript (Fig 3C, D and EV3A, B) to enhance the robustness and reliability of the results.

9. It is known that excessive intracellular lactate reduces glycolytic flux 17, therefore the high level of lactate in the mitotic cells might be likely from extracellular lactate. How does this make sense? Lactate imported from the extracellular environment will also reduce the

glycolytic flux, so there is essentially no reason for cells to import it unless they are going to use it as a carbon source, otherwise it will block glycolysis.

Response: We thank the reviewer for pointing out the mistake. We have revised and clarified the statement as follows: “Since pyruvate conversion to lactate (glycolytic flux) is slightly reduced in mitosis (**Fig. 1E**), the observed high lactate levels likely originate from extracellular sources.”

10. Without AZD3965, addition of exogenous lactate led to a twofold increase of cytosolic lactate level during mitosis, which accelerated mitotic progression. I am not sure how the authors can talk about cytosolic lactate levels in mitosis, as there is no nucleus in mitosis.

Response: We thank the reviewer for pointing out the mistake. We have revised the term “cytosolic lactate” to “intracellular lactate” in the manuscript for accuracy. Our results are consistent with previous findings (Nature. 2023. PMID: 36921622, **Fig. 4a**), which also reported an increase in lactate levels during mitosis.

11. How generalizable is the system? What happens in cells that do not express LDHB, such as T47D cells? What happens in cells that express low levels of MCT1, such as MDAMB231? (Other cell line expressing low/high levels of either LDHs or MCTs can be found in the CCLE proteomics database)

Response: We thank the reviewer for raising these critical questions about the system’s generalizability. Our functional data support that both LDHB and MCT1 are required for lactate-dependent mitotic progression. In LDHB-deficient cells (T47D) and cells with low MCT1 expression (MDA-MB-231), exogenous lactate supplementation failed to enhance mitotic progression (**Fig. EV3H**). This indicates that the presence of both LDHB and MCT1 is essential for this mechanism.

To assess broader relevance, we examined LDHA T18 phosphorylation—a key readout of our proposed pathway—across 14 cell lines that expressing LDHA/B and MCT1 (**Fig. EV4D**). High phosphorylation levels were observed in 12/12 cancer cell lines tested, while two non-transformed lines showed low levels. This suggests that CDK1-mediated LDHA T18 phosphorylation is a conserved feature in cancer cells.

However, while the phosphorylation event is widespread in cancer, these data do not definitively prove that the functional consequence (i.e., lactate-dependent modulation of LDH tetramer composition to support mitosis) operates identically in all contexts. As noted by the reviewer, proteomic heterogeneity across cell lines (e.g., variable LDHA/LDHB/MCT1 ratios) may influence pathway dynamics.

Therefore, while the core mechanism (CDK1-LDHA T18 phosphorylation) is prevalent in cancer cells, The functional output (lactate-driven mitotic support) requires co-expression of LDHA/B and MCT1, as demonstrated in our loss-of-function models (**Fig. EV3H**). Future studies with metabolic profiling in diverse cell types are needed to fully map the system's universality. We have added this limitation into the Discussion.

12. It is not clear how the authors monitor lactate-mediated mitotic effects: how long after lactate treatment are cells followed in live cell imaging? How many cells per biological replicate? How are cells selected? all at mitotic entry? why did the author not arrest cells in G2 to facilitate analysis?

Response: Following careful lactate supplementation, live-cell imaging was initiated within approximately one to two minute to minimize metabolic alterations. Only cells undergoing unperturbed mitosis—not those released from arrest—were analyzed to preserve their physiological metabolic state. Cells in mitosis were identified based on characteristic rounded morphology and chromosome alignment, with a focus on metaphase cells exhibiting properly aligned chromosomes. Selection was performed without bias.

For Figure 3C, using $n = 6$ biologically independent cells for both interphase and mitotic conditions.

For Figure 3F, mitotic duration was quantified from the onset of nuclear envelope breakdown to the point of chromosome segregation, a total of $n = 93, 92, 93, 92, 91$ and 93 biologically independent cells were analyzed across the respective experimental conditions.

13. Is there any evidence that MCT1 transporters are regulated during mitosis? Are they cell cycle dependent genes? Can they be found in the 10.7554/eLife.01630 dataset? how do the authors explain that the expression of MCT1 increased in the 1 hour duration of mitosis?

Response: We thank the reviewer for raising these critical questions regarding MCT1 regulation during mitosis. Our data and published literature collectively demonstrate that MCT1 is translationally regulated in a cell cycle-dependent manner:

We demonstrated MCT1 protein abundance significantly increased within 1 hour of mitotic entry (Fig. EV3D). Extended mitotic arrest (2 h MG132 treatment) did not further elevate MCT1 levels, indicating rapid and transient translation. Inhibition of protein synthesis by cycloheximide (CHX) abolished this increase (Fig. EV3E), confirming de novo translation drives mitotic MCT1 accumulation.

Our data is consistency with prior studies. Molecular Cell (PMID: 24120665) reported translational upregulation of SLC16A1 (MCT1) during mitosis, aligning with our mechanistic conclusions.

Elife (PMID: 29052541) showed progressive MCT1 accumulation across G1/S/G2/M phases (Supplementary File 2), supporting its broad cell cycle dependence.

Gene	Symbol	G1		S-phase		Mitosis	
		Q-value	Change	Q-value	Change	Q-value	Change
ENSG00000155380.7	SLC16A1	0.063424	N.S.	0.002656	Up	0.002892	Up
ENSG00000134057.10	CCNB1	0.001024	Up	0.130564	N.S.	0.006952	Up
ENSG00000170312.10	CDK1	0.092908	N.S.	0.110883	N.S.	1.44E-10	Up
ENSG00000166851.8	PLK1	0.000417	Up	0.025646	N.S.	0.008255	Up

Molecular Cell. 2013. PMID: 24120665

Protein.names	Gene.names	G1.median	S.median	G2.median	M.median
Monocarboxylate transporter 1	SLC16A1	1.1389	1.5362	1.8330	1.9176
Mitotic checkpoint protein BUB3	BUB3	1.1389	1.5590	2.0454	2.2607
Kinesin-like protein KIF2C	KIF2C	1.1389	1.5476	2.3427	2.8490
Cyclin-dependent kinase 1	CDK1	1.1389	1.4241	1.9074	2.8779
Serine/threonine-protein kinase PLK1	PLK1	1.1389	1.2014	3.2795	4.7754
G2/mitotic-specific cyclin-B1	CCNB1	1.1389	2.9133	5.3234	7.7347

Elife. 2017. PMID: 29052541

In the dataset from 10.7554/eLife.01630, titled "A proteomic chronology of gene expression through the cell cycle in human myeloid leukemia cells", MCT1 is not on the list.

Conclusion: MCT1 expression is cell cycle-regulated, with mitotic upregulation mediated by transient translational activation—a mechanism conserved across multiple cancer types.

14. In the images shown in Figure 4, LDHA pT18 appears to localise to the mitotic spindle and centrosomes, supporting a role in mitosis beyond the indirect role attributed to it by the authors. Does LDHAT18A cause lagging chromosomes due to mislocalisation or disruption of its role in the mitotic spindle? Does the mitotic spindle show defects in cells expressing LDHAT18A?

Response: We appreciate the reviewer for raising this important point. We also observed that LDHA pT18 exhibits elevated localization on the spindle and spindle poles. This distribution may be attributed to the presence of CDK1/cyclin B, which is known to localize to the spindle and centrosomes during mitosis and could potentially mediate site-specific phosphorylation of LDHA. (reference: J Cell Biol. 2010. PMID: 20404109; Cell Res. 2008. PMID: 18195732; Cell Motil Cytoskeleton. 1990. PMID: 2268875).

15. CDK1-mediated phosphorylation of LDHA is required for mitotic progression and tumor growth -> this cannot be correct because LDHA repression does not have any mitotic defects, according to the authors

Response: Thank you for raising this important point. We acknowledge that this was not clearly explained in the previous version. LDHA depletion or CDK1-mediated phosphorylation of LDHA promotes the formation of LDH tetramers with increased incorporation of LDHB, which supports normal mitotic progression. In contrast, expression of the non-phosphorylatable LDHA T18A mutant reduces the proportion of LDHB in the LDH tetramer, leading to decreased ATP production and resulting in mitotic defects.

16. 5C, colours should be shown in individual images. How many cells per paired sample were quantified? This is not specified, as is generally the case with many figures. The statistical test is also not mentioned.

Response: We thank the reviewer for highlighting these essential methodological details. We analyzed 15 paired clinical samples (colorectal cancer vs. adjacent normal tissue) with total 31 mitotic cells (average: 2.1 cells/sample; range: 1-3 cells per paired set). Individual data points are shown for all 31 cells, and paired measurements (tumor vs. normal from same patient) are connected by lines and analyzed by paired two-tailed Student's t-test. These details have been

added to the **Fig. 5D** legend in the revised manuscript

17. 5B: How many mitotic cells have been analysed per replicate? Where are the source data?

Response: For each replicate, at least 15–20 mitotic cells were analyzed. The corresponding source data have been provided in the respective figure folder.

18. A recent publication in Nature showed that lactate controls the APC: the authors barely mention this in the discussion. How does their work relate to this publication? Are these two unrelated mechanisms by which lactate controls mitosis?

Response: We have thoroughly compared our findings with those of this study in the Discussion section of the revised manuscript. The contents are as follows.

Recent study discovered a non-metabolic function of lactate as a receptor activation in mitosis (Liu et al, 2023). Notably, Liu et al demonstrated that lactate accumulation binds and inhibits deSUMOylating enzyme SENP1, stabilizing the APC/C complex to drive mitotic exit independently of LDH enzymatic activity (Liu et al., 2023). In contrast, our work establishes that lactate also promotes mitotic progression through its metabolic conversion via LDHB to generate NADH/ATP. These parallel yet independent pathways illustrate lactate's multifaceted roles in mitosis, likely operating in a cell type- or context-dependent manner.

19. Does inhibiting the ETC have an effect on mitotic progression? Would it be different to inhibit the ETC at different levels? Does this regulation of lactate in mitosis change depending on the glycolytic state of the cells? To test whether ATP is the molecule required for mitosis progression and its dependence on NADH, the authors should treat cells with either oligomycin or BAM15. The latter is a mitochondrial uncoupler that increases electron transfer but slows down ATP synthesis.

Response: To investigate the role of mitochondrial function in mitosis, we employed both glycolytic (HeLa, HCT116) and oxidative (SiHa, U2OS) cancer cell lines for ETC (electron transport chain) inhibition studies. Inhibition of the ETC resulted in prolonged mitotic duration and a higher incidence of lagging chromosomes. Furthermore, increasing the concentration of ETC inhibitors exacerbated these mitotic defects in a dose-dependent manner.

To validate the influence of the glycolytic state on the regulatory effects of lactate during mitosis, we specifically utilized both glycolytic and oxidative cancer cell types (Fig. EV3F, 3G). In agreement with previous findings (J Clin Invest. 2008; PMID: 19033663) showing elevated lactate uptake in oxidative cancer cells, the mitosis-promoting effect of lactate was more prominent in oxidative tumor cells. Notably, supplementation of SiHa cells with exogenous lactate reduced mitotic duration by approximately 14 minutes.

Additionally, previous studies have shown that treatment with oligomycin not only depletes ATP but also elevates reactive oxygen species (ROS) levels (Rheumatology (Oxford). 2014. PMID: 24609059; BMC Musculoskelet Disord. 2017. PMID: 28606072). In our experiments, co-treatment with ATP and the antioxidant N-acetylcysteine (NAC) partially rescued the mitotic defects induced by oligomycin, further supporting the dual contribution of energy and redox homeostasis in mitotic regulation.

20. ATP is above 1 mM in the cells. This is considered excessive for cellular requirements. What is the metabolic demand for ATP in mitotic cells that would justify a requirement for this shift in LDH complexes?

Response: We thank the reviewer for this insightful question regarding ATP homeostasis in mitosis. The requirement for elevated ATP during mitosis is driven by two critical demands: 1. Energy-intensive mitotic processes: Chromosome segregation, spindle dynamics, and checkpoint signaling consume ATP at rates exceeding interphase levels (Trends Cell Biol. 2017. PMID: 27746095; PNAS. 2010. PMID: 20212161; PNAS. 2011. PMID: 21300909). Mitotic kinases (Aurora A/B, Bub1) and chromosomal regulators (INCENP) require millimolar ATP not only as a substrate but as a structural stabilizer for complex assembly (Nat Commun. 2019. PMID: 30858367). 2. ATP-dependent biophysical regulation: Thermal profiling reveals ATP concentrations <500 μ M support canonical kinase activity, while 1-2 mM ATP is necessary to maintain solubility of phase-separated mitotic complexes (Nat Commun. 2019. PMID: 30858367).

This heightened ATP demand (Trends Cell Biol. 2017. PMID: 27746095; Cell Rep. 2024. PMID: 39342616; Cell Rep. 2021. PMID: 34010649; among others) creates selective pressure for metabolic adaptations. Our data demonstrate that CDK1-mediated LDHA T18 phosphorylation: Favors LDHB enriched tetramers to generate NADH and subsequent ATP. Thus, LDH complex remodeling represents a rapid metabolic adaptation to power ATP-dependent structural and enzymatic processes unique to mitosis.

Effect of ATP depletion of proteome thermal stability			
protein_id	gene_name	$\Delta Tm(D1), ^\circ C$	$\Delta Tm(D2), ^\circ C$
O14965	AURKA	-2.5015	-1.9439
Q96GD4	AURKB	-1.0449	-1.3447
Q9NQS7	INCENP	-0.7317	-1.6712
O43683	BUB1	-2.9624	-2.1185
Q12834	CDC20	-1.1314	-0.6509
P53350	PLK1	-0.3316	-1.2446

Proteome solubility upon ATP depletion			
protein_id	gene_name	SDS(D1) log2(D1/control)	SDS(D2) log2(D2/control)
O14965	AURKA	0.0120	0.0226
Q96GD4	AURKB	0.1116	0.1719
Q9NQS7	INCENP	0.0682	0.1677
O43683	BUB1	0.0342	0.0761
Q12834	CDC20	0.0962	0.0434
P53350	PLK1	0.0226	0.0298

Nat Commun. 2019. PMID: 30858367

In general, I would recommend looking out for oversimplified conclusions, lack of statistical detail, lack of source data, lack of important experimental details such as number of biological replicates or cells analysed.

Reviewer #3 (Remarks to the Author):

Overall, the authors do provide a clear and interesting advancement regarding our understanding of mitotic metabolism. Little is known about the regulation of lactate metabolism in mitosis and the results presented here will be of interest to the field. However, the manuscript has some major issues, as some of the conclusions are not justified by the data, some key controls are missing, and the results are not discussed in relation to existing literature. Correcting these issues is essential before this work can be published. However, addressing these issues should be relatively easy and doing so will increase the impact of the work, while also making the manuscript more transparent. Please see below for details.

Response: We are grateful for the reviewer's positive comments regarding the overall quality and significance of our work. We appreciate the thoughtful feedback and encouragement.

Major concerns:

1. Conclusions about mitochondrial metabolism are not justified by the data

Across the manuscript (title, abstract, intro, results, several sections in the discussion, and Fig 5D) the authors imply that the metabolic changes they identify in mitosis will result in increased ATP synthesis that supports mitosis. Not only is this conclusion not justified by the data, but it's also questionable considering existing literature. The authors show no data on mitochondrial metabolic rates, not to mention ATP synthesis rates in mitochondria or glycolysis. The fact that ATP levels decrease in mitosis does not suffice to conclude about the metabolic fluxes (synthesis rates).

The reported decrease in ATP levels (~30% decrease) is unlikely to influence any mitosis driving protein activity, as cellular ATP levels are over an order of magnitude higher than the K_m values of motor proteins etc. The fact that ATP supplementation rescues some of the mitotic defects observed by the authors is curious, but this does not prove that the changes in lactate metabolism drive ATP synthesis.

To address this, I think the authors have two options. They could carry out extensive metabolite tracing experiments and oligomycin-responsive oxygen consumption experiments to test how ATP synthesis rates change in mitosis. The authors should also be able to rescue the mitotic defects caused by LDHB KD by driving more NADH into mitochondria (e.g. via MDH1 and GPD1L overexpression). However, considering that the manuscript's key discoveries (CDK1 controlling lactate metabolism) are not reliant on these conclusions about ATP synthesis and mitochondrial metabolism, the authors could just remove all these claims from the manuscript. The work would still be sufficiently interesting for publication in EMBO reports, and the work might be an even better fit for the journal's requirement that "conclusions constitute a single key message worthy of publication in EMBO reports".

Response: We appreciate the reviewer's thoughtful summary of our study and their insightful evaluation of both the strengths and limitations of our data. We fully agree that the central finding of this study is the regulation of LDHA by CDK1 and the functional role of LDHB in mitosis. As noted by the reviewer, our current evidence on ATP generation from lactate-derived NADH is limited. In response, we have revised the relevant claims in the Results section to better reflect this limitation.

Additionally, we have expanded our discussion of this issue in the Discussion section. Corresponding revisions have also been made to the related statements and the working model to more accurately represent the proposed mechanism.

2. Key experimental controls are missing and reproducibility is unclear:

- The authors need to validate the specificity of the antibody they've generated to detect phosphorylated LDHA, and these validations need to be shown in the supplements. The authors need to compare control and LDHA KD cells. I would expect to see IF images of these cells labelled with the antibody along with quantifications of the signal intensities.

Response: In accordance with the reviewer's suggestion, we have thoroughly validated the specificity of the LDHA pT18 antibody. As shown in **Figure 4D** of the revised manuscript, LDHA pT18 signals were compared between wild-type and LDHA knockdown cells, demonstrating reduced signal upon LDHA depletion. Additionally, pre-immune IgG controls were tested and included in **Figure EV4C** to further confirm antibody specificity.

3. I would also expect to see a full western blot membrane following labeling with this antibody. In addition, there are currently no method details about the antibody generation or validation, and this needs to be corrected.

Response: The full western blot membrane probed with the LDHA pT18 antibody has been included in the revised manuscript (**Fig. 4E**). Additionally, detailed information on the generation and validation of the antibody has been provided in the Methods section to ensure transparency and reproducibility.

4. Based on the methods section, most experiments are done by comparing G1/S synchronized cells to M-phase synchronized cells. I'm glad to see that the authors used mitotic shake-off for the enrichment of M-phase cells, as this is likely to yield more 'normal' mitotic cells than mitotic arrests. However, key experiments should include a comparison to unsynchronized cells and/or an alternative interphase enrichment method (e.g. leftovers from the mitotic shake-off). Otherwise, the conclusions could be due to changes specifically in G1/S transition rather than M-phase.

Response: We sincerely appreciate the reviewer's agreement with our cell cycle synchronization methodology. In accordance with the reviewer's recommendation, we have included an unsynchronized cell population—comprised of over 90% interphase cells—as an internal control. This addition allows for a direct comparison of MCT1/4 expression levels across synchronized interphase cells, mitotic cells, and asynchronous populations, as shown in **Figure 3E** of the revised manuscript. Moreover, **Figure 4E** also included an unsynchronized cell population.

5. In addition, the authors need to clarify when were cells synchronized to mitosis using mitotic arrest (as done in Fig 4E), and how exactly this was done.

Response: In Figure 4E and 4F, to synchronize cells in mitosis, we employed a double thymidine block and release assay. Once the cells progressed into mitosis, they were treated with 10 μ M MG132 for 30 minutes to arrest them in metaphase, followed by the addition of various inhibitors for an additional 30 minutes. Cells were then either collected for western blot analysis or fixed for immunofluorescence staining. This methodology has been clearly described and clarified in the revised manuscript.

6. The consistency of LDHA and LDHB knockdowns is unclear. In Fig S1B, the LDHB protein band disappears with the KD, but then reappears partly when Flag-rLDHB is introduced (i.e. the smaller band on the blot is probably endogenous LDHB). This suggests that the KD is not always complete nor consistent. The authors should quantify the KD efficiency across experiments and cell lines and show this in the supplements.

Response: Knockdown efficiency has been quantified and is now included in the revised manuscript (Fig. EV1A) to support the validity of the RNAi experiments.

7. Fig 1F is critical for this paper and its key conclusions. While the blot shown is convincing, the reproducibility of this finding is unclear. How many times has this experiment been replicated? Please show quantifications across experiments and, preferably, in different cell lines too.

Response: In the revised manuscript, we have quantified the ratios of LDH isoenzymes based on data from three independent experiments. These results are now presented in Figure 1F to support the reproducibility and robustness of our findings.

8. Similarly, additional data from a different cell line would strengthen the Fig 1E, which is also critical to this manuscript.

Response: We have also measured LDH enzymatic activity in DLD1 cells, and the results are included in the revised manuscript (Fig. 1E) to further support our findings.

9. Optimally, the reverse activity of the LDH isoenzyme would be verified with metabolite tracing experiments. This would significantly increase the credibility of this work. However, such assays may not be easily available to the authors, and if this validation cannot be done the lack of it should be acknowledged in the text.

Response: We appreciate the reviewer's thoughtful suggestion and have acknowledged this point in the text of the revised manuscript accordingly.

10. Fig 1 is missing some controls. What is the typical mitotic behavior of WT cells in Fig 1B? How efficient is the LDHB knockdown in Fig 1C when working with organoids?

Response: We have included two control groups in our analysis, as shown in **Figure 1A**. Additionally, knockdown efficiency in organoids was confirmed by western blotting and is presented in **Figure 1C** of the revised manuscript.

11. The MDH1 and GPD1L are knocked out, but the western blot in 2F shows weak bands of the right size. This should not happen if the KO is successful. There are no method details on how the KO was carried out. The KOs need to be validated and described properly.

Response: Knockdown cells were stringently selected using puromycin. Knockout efficiency was confirmed by western blotting, as shown in **Figure EV2D**. Detailed methods for the knockout procedure have been included in the Methods section of the revised manuscript.

12. The results of fluorescence-based metabolite sensors need clarification. In Fig 3C,D, how many cells were examined and in how many experiments? Are the sensor responses linear with the metabolite change? If not, the comparison between interphase and mitotic cells using the sensors (Fig 3C,D) seem questionable.

Response: We have added quantified results ($n = 6$) for these experiments, as shown in **Figure 3C** and **3D**. Although the sensor responses are not strictly linear across all concentration ranges, the readouts remain positively correlated with lactate concentration and the NAD^+/NADH ratio. This is consistent with previously published studies utilizing similar sensors (SoNar. Cell Metabolism. 2015. PMID: 25955212; FiLa. Cell Metabolism. 2023. PMID: 36309010)

Fila, Cell Metabolism 2023

SoNar, Cell Metabolism 2015

13. In Fig 3G, the authors should consider also showing how intracellular lactate levels change with the MCT knockdowns.

Response: We have included this data in the revised manuscript (Fig. 3G).

14. The in silico analyses in Fig 4J,K require controls that allow the evaluation of effect specificity and magnitude. Would phosphorylation on any phospho-site result in similar effects?

Response: In the revised manuscript, we have included LDHA Y10 phosphorylation (Mol Cell Biol. 2011. PMID: 21969607; Oncogene. 2017. PMID: 28218905) as a control. In contrast to T18

phosphorylation, Y10 phosphorylation did not result in significant dynamic changes in RMSD over time, as shown in **Figure EV4E**.

15. Existing literature is ignored, which influences the interpretation of the data

There are several publications from recent years that are not acknowledged by the authors, even though these publications could help interpret the authors' work. This limits the impact of the work and makes some aspects of the writing seem controversial.

- Mitochondrial ETC activity seems to be increased in mitosis, but ATP synthesis may decrease because mitochondrial ATP synthase is partially inhibited (1)
- Glycolytic fluxes may be directed away from pyruvate in mitosis (2)
- ATP level decrease in mitosis is well documented (1, 3, 4)
- The authors also suggest that biosynthesis rates in mitosis are low and energy is used for cell division. While there's no clear evidence that cell division requires a lot of energy, there are many papers showing that biosynthesis remains nearly at interphase levels during normal mitosis (when cells are not arrested in mitosis) (5-7)

Response: We thank the reviewer for raising this important issue. We agree that the energy metabolism of mitotic cancer cells is likely context-dependent, potentially influenced by the glycolytic status and mitochondrial function of individual cell lines. As a result, metabolic behavior may vary across different cancer types. In the revised manuscript, we have discussed these considerations and cited relevant publications to support this context-specific interpretation.

16. Is the LDHA T18 phosphorylation typical in normal cells?

In Fig 5C, there is little LDHA T18 phosphorylation in the non-tumor tissue. This should be acknowledged and discussed, as it seems like this phosphorylation may not be feature of normal, non-cancerous cells. Optimally, the authors would test the LDHA T18 phosphorylation levels in several normal and cancerous models (either in vivo or just cell lines), but at minimum this needs to be acknowledged in the text.

Response: In the revised manuscript, we analyzed LDHA T18 phosphorylation levels across 14 cell lines expressing LDHA/B and MCT1 (**Fig. EV4D**). We found that LDHA T18p levels are markedly elevated in most cancer cell lines, whereas much lower levels were observed in non-transformed cells such as MCF10A and RPE. Furthermore, LDHA T18 phosphorylation

levels in mitotic cells were substantially higher in CRC tumor tissues than in paracancerous tissues (**Fig. 5D**). These findings suggest that LDHA T18 phosphorylation is a conserved feature in cancer cells. This observation has been discussed in detail in the Discussion section.

Minor points:

17. I recommend noting in the results section how the cells were synchronized, as this is important for the experimental outcome.

Response: We have described the cell synchronization method in the Results section of the revised manuscript to clarify the experimental approach.

18. In figure legends, the authors use the expression "biologically independent experiments/samples". This should be clarified somewhere. Were these independent cultures grown and measured at the same time? Or were these separate experiments from different days? And is there a difference between experiments and samples when detailing these?

Response: In the revised manuscript, we have clarified the details of the statistical analyses in the Methods section. Additionally, we have specified relevant statistical information in the corresponding figure legends to enhance transparency and reproducibility.

19. The manuscript could be shortened by moving Figures 1A, 2F, 4A, 4J, 4K to the supplement. Also the discussion could be cut shorter by removing repetitive detailing of the findings and by cutting away parts of the second to last paragraph.

Response: We agree with the reviewer's suggestion and have moved the relevant figures to the Extended View section in the revised manuscript.

20. How did the authors select the cancer types to evaluate in Fig 1A? Are there potential sampling biases?

Response: We thank the reviewer for raising this issue. We selected tissue samples and cell lines from the TCGA dataset and several published studies based on high expression levels of both LDHA and LDHB (PMID: 38291366; PMID: 38709280), and have cited these references in the revised manuscript. However, in accordance with the suggestion from the other Reviewers, these data have been removed to streamline the manuscript and maintain focus.

21. When describing Fig 1E results in the text, authors claim 2-fold increase in lactate to pyruvate conversion rates. This is an overstatement based on the data figure.

Response: Thank you for pointing out the mistake. We have corrected the description in the revised manuscript accordingly

22. The results could be informative of why cancer cells undergo the Warburg effect. This seems completely ignored in the discussion section.

Response: Thank you for the constructive suggestion. We have incorporated this topic into the Discussion section of the revised manuscript to address its relevance and implications.

References:

1. J. H. Kang, et al., Monitoring and modeling of lymphocytic leukemia cell bioenergetics reveals decreased ATP synthesis during cell division. *Nat Commun* 11, 4983 (2020).
2. X. Ma, et al., Polo-like kinase 1 coordinates biosynthesis during cell cycle progression by directly activating pentose phosphate pathway. *Nat Commun* 8 (2017).
3. K. Maeshima, et al., A transient rise in free Mg (2+) ions released from ATP-Mg hydrolysis contributes to mitotic chromosome condensation. *Current Biology* 28, 444-451 (2018).
4. E. Doménech, et al., AMPK and PFKFB3 mediate glycolysis and survival in response to mitophagy during mitotic arrest. *Nat Cell Biol* 17, 1304-1316 (2015).
5. T. P. Miettinen, J. H. Kang, L. F. Yang, S. R. Manalis, Mammalian cell growth dynamics in mitosis. *Elife* 8 (2019).
6. V. Stonyte, E. Boye, B. Grallert, Regulation of global translation during the cell cycle. *J Cell Sci* 131 (2018).
7. M. Shuda, et al., CDK1 substitutes for mTOR kinase to activate mitotic cap-dependent protein translation. *Proc Natl Acad Sci U S A* 112, 5875-5882 (2015).

Dear Dr. Yang

Thank you for the submission of your revised manuscript to our editorial offices. I have now received the reports from the three referees that I asked to re-evaluate the study, you will find below. As you will see, the referees now support the publication of your study in EMBO reports. Referees #2 and #3 have further comments and suggestions to improve the manuscript, I ask you to address in a final revised manuscript. Please also provide a final p-b-p-response to these points and my editorial requests below.

Editorial requests:

- Please provide a more comprehensive final manuscript title with not more than 100 characters including spaces and without typos.
- The resolution of the submitted figures is too low for blots and microscopy images. This reduction in resolution is commonly caused by converting original 16-bit TIFF files to RGB format for publication. While this is not inherently problematic, it can raise concerns about image integrity for critical readers.

To avoid any misunderstanding and to meet EMBO Press standards, we kindly ask that you:

- * Resubmit the complete figure set at the captured original data resolution.
 - * Apply the same resolution standards to the blot source data files, which are also currently below the required quality threshold. Please upload the blot source data files as .tiff or pdf. Please do not use PowerPoint.
 - Please also update the source data checklist regarding panels moved to one of the EV figures.
 - As your manuscript contains just 5 final main and 5 final EV figures, we will publish your manuscript as Report. For a Scientific Report we require that results and discussion sections are combined in a single chapter called "Results & Discussion". Please do this for your manuscript. For more details please refer to our guide to authors:
<http://www.embopress.org/page/journal/14693178/authorguide#researcharticleguide>
 - Please add scale bars of similar style and thickness to microscopic images, using clearly visible black or white bars (depending on the background). Please place these in the lower right corner of the images themselves. Please do not write on or near the bars in the image but define the size in the respective figure legend. Presently, some of the scale bars are rather thin and hard to see, or sit in the upper part of the image. Please improve.
 - Please check again that the number "n" for how many independent experiments were performed, their nature (biological versus technical replicates), the bars and error bars (e.g. SEM, SD) and the test used to calculate p-values is indicated in the respective figure legends. Please also check that all the p-values are explained in the legend, and that these fit to those shown in the figure. Please provide statistical testing where applicable. Please avoid the phrase 'independent experiment' but clearly state if these were biological or technical replicates. Please also indicate (e.g. with n.s.) if testing was performed, but the differences are not significant. In case n=2, please show the data as separate datapoints without error bars and statistics. See also:
<http://www.embopress.org/page/journal/14693178/authorguide#statisticalanalysis>
- If n<5, please show single datapoints for diagrams. Moreover:
- Please provide exact p values in the legends of figures 1C, E; 2 B, C, D; 3B, C, D; 4D, F, G, I; 5A, B; EV1 A, B, C, D; EV2 C, EV3 C, EV4 C
 - Please add to each legend (main, EV figures and Appendix Figures, where applicable) a 'Data Information' section (or name the provided 'notes' section like this) explaining the statistics used or providing information regarding replicates and scales. See:
<https://www.embopress.org/page/journal/14693178/authorguide#figureformat>
 - The data availability section (DAS) is restricted to externally deposited large datasets generated in a study. If no primary datasets have been deposited, please state this in this section ("No large primary datasets have been generated and deposited for this study"). Please remove all other information from this section.
 - Please make sure that all the funding information is also entered into the online submission system and that it is complete and similar to the one in the acknowledgement section of the manuscript text file. Presently, these grants are presently missing from the submission system: The Strategic Priority Research Program of the Chinese Academy of Sciences, Grant XDB0940101; National Key R&D Program of China (2022YFA1303100); the Fundamental Research Funds for the Central Universities

(YD9100002028); Center for Advanced Interdisciplinary Science and Biomedicine of IHM of USTC (QYPY20220017); the "Laboratory for Synthetic Chemistry and Chemical Biology" under the Health@InnoHK Program by the Innovation and Technology Commission of Hong Kong.

In addition, I would need from you uploaded separately:

Best,

Referee #1:

Thank you for being so responsive and open-minded to my suggestions and critiques. I have no further issues with this manuscript. Congratulations on this amazing work, and I wish you all the best.

Referee #2:

I commend the authors for the revision, which significantly strengthens the findings. I am satisfied with the changes provided. However, I would suggest the authors revise their proteomics analysis of LDHB and LDHA. Despite their homology, these two enzymes can be reliably distinguished in mass spectrometry experiments. Their peptide fragmentation patterns do not produce identical peptides, allowing them to be specifically recognized in the analysis.

Referee #3:

The revised manuscript by Liu, Cheng, et al. provides large amounts of novel data on mitotic metabolism. The revisions to the manuscript were extensive and they addressed most of the technical concerns I had with this paper.

The manuscript is not flawless. There are some important inconsistencies between this work and those published before, especially regarding ATP levels and production/consumption in mitosis. In addition, the mechanism of some of the rescue experiments, such as the NADH supplementation, remain unclear (it's not clear if/how external NADH is taken up by the cell). Also, the writing of the discussion section could be improved to avoid partly contradicting statements, especially when the published literature is not in full agreement with the results shown, and to more clearly state what conclusions are partly speculative.

Despite these shortcomings, I believe this manuscript will be of great interest to the research field.

We sincerely thank the Editor and Reviewers for their positive comments and constructive suggestions. Below, we provide detailed point-by-point responses (in blue text) to each concern raised.

Referee #1:

Thank you for being so responsive and open-minded to my suggestions and critiques. I have no further issues with this manuscript. Congratulations on this amazing work, and I wish you all the best.

Response: We thank the reviewer for acknowledging our research.

Referee #2:

I commend the authors for the revision, which significantly strengthens the findings. I am satisfied with the changes provided.

However, I would suggest the authors revise their proteomics analysis of LDHB and LDHA. Despite their homology, these two enzymes can be reliably distinguished in mass spectrometry experiments. Their peptide fragmentation patterns do not produce identical peptides, allowing them to be specifically recognized in the analysis.

Response: We thank the reviewer for the corrections and suggestions. Following re-analysis of the mass spectrometry data on LDH subunit expression using the specified database, we still observed significant variation in LDHA and LDHB expression across tumor types. To prevent potential misinterpretation, we have omitted these analyses from the revised manuscript. Additionally, we have now emphasized in the Abstract, Results, and Discussion that our findings specifically apply to cancer cells co-expressing both LDHA and LDHB.

Referee #3:

The revised manuscript by Liu, Cheng, et al. provides large amounts of novel data on mitotic metabolism. The revisions to the manuscript were extensive and they addressed most of the technical concerns I had with this paper.

The manuscript is not flawless. There are some important inconsistencies between this work and those published before, especially regarding ATP levels and production/consumption in mitosis. In addition, the mechanism of some of the rescue experiments, such as the NADH supplementation, remain unclear (it's not clear if/how external NADH is taken up by the cell). Also, the writing of the discussion section

could be improved to avoid partly contradicting statements, especially when the published literature is not in full agreement with the results shown, and to more clearly state what conclusions are partly speculative.

Despite these shortcomings, I believe this manuscript will be of great interest to the research field.

Response: We thank the positive comments from the reviewer. We also appreciate the reviewer for reminding us of the limitations in our work and for the suggestion to revise the discussion. In the Results section, we have added the statement: "The intracellular NADH/NAD⁺ ratio increased rapidly following extracellular NADH addition (Fig. 2F), although the mechanism of NADH entry remains unclear." Additionally, we have revised the Discussion to explicitly address the limitations of our findings.

Editorial requests:

1. Please provide a more comprehensive final manuscript title with not more than 100 characters including spaces and without typos.

Response: The revised title now reads: "CDK1-Mediated Phosphorylation of LDHA Fuels Mitosis through LDHB-Dependent Lactate Oxidation."

2. The resolution of the submitted figures is too low for blots and microscopy images. This reduction in resolution is commonly caused by converting original 16-bit TIFF files to RGB format for publication. While this is not inherently problematic, it can raise concerns about image integrity for critical readers.

To avoid any misunderstanding and to meet EMBO Press standards, we kindly ask that you:

* Resubmit the complete figure set at the captured original data resolution.

* Apply the same resolution standards to the blot source data files, which are also currently below the required quality threshold. Please upload the blot source data files as .tiff or pdf. Please do not use PowerPoint.

Response: We have now resubmitted the original figures as TIFF files and uploaded the source blot files as PDFs.

3. Please also update the source data checklist regarding panels moved to one of the EV figures.

Response: We have updated the source data checklist.

4. As your manuscript contains just 5 final main and 5 final EV figures, we will

publish your manuscript as Report. For a Scientific Report we require that results and discussion sections are combined in a single chapter called "Results & Discussion". Please do this for your manuscript. For more details please refer to our guide to authors:

<http://www.embopress.org/page/journal/14693178/authorguide#researcharticleguide>

Response: We have revised the section as requested.

5. Please add scale bars of similar style and thickness to microscopic images, using clearly visible black or white bars (depending on the background). Please place these in the lower right corner of the images themselves. Please do not write on or near the bars in the image but define the size in the respective figure legend. Presently, some of the scale bars are rather thin and hard to see, or sit in the upper part of the image. Please improve.

Response: We have revised scale bars accordingly.

6. Please check again that the number "n" for how many independent experiments were performed, their nature (biological versus technical replicates), the bars and error bars (e.g. SEM, SD) and the test used to calculate p-values is indicated in the respective figure legends. Please also check that all the p-values are explained in the legend, and that these fit to those shown in the figure. Please provide statistical testing where applicable. Please avoid the phrase 'independent experiment' but clearly state if these were biological or technical replicates. Please also indicate (e.g. with n.s.) if testing was performed, but the differences are not significant. In case n=2, please show the data as separate datapoints without error bars and statistics. See also:

<http://www.embopress.org/page/journal/14693178/authorguide#statisticalanalysis>

If n<5, please show single datapoints for diagrams. Moreover:

- Please provide exact p values in the legends of figures 1C, E; 2 B, C, D; 3B, C, D; 4D, F, G, I; 5A, B; EV1 A, B, C, D; EV2 C, EV3 C, EV4 C

Response: We have provided exact p values in the legends of each figure.

7. Please add to each legend (main, EV figures and Appendix Figures, where applicable) a 'Data Information' section (or name the provided 'notes' section like this) explaining the statistics used or providing information regarding replicates and scales. See:

Response: We have added the 'Data Information' section to each legend.

8. The data availability section (DAS) is restricted to externally deposited large datasets generated in a study. If no primary datasets have been deposited, please state this in this section ("No large primary datasets have been generated and deposited for this study"). Please remove all other information from this section.

Response: We have streamlined the Data Availability Statement (DAS) by removing non-essential content and explicitly stating: "No large-scale primary datasets were generated or deposited in this study."

9. Please make sure that all the funding information is also entered into the online submission system and that it is complete and similar to the one in the acknowledgement section of the manuscript text file. Presently, these grants are presently missing from the submission system: The Strategic Priority Research Program of the Chinese Academy of Sciences, Grant XDB0940101; National Key R&D Program of China (2022YFA1303100); the Fundamental Research Funds for the Central Universities (YD9100002028); Center for Advanced Interdisciplinary Science and Biomedicine of IHM of USTC (QYPY20220017); the "Laboratory for Synthetic Chemistry and Chemical Biology" under the Health@InnoHK Program by the Innovation and Technology Commission of Hong Kong.

Response: We have added these grants information into the online system.

10. In addition, I would need from you uploaded separately:

Response: We have uploaded separately the summary, key points and a schematic summary figure.

Dr. Zhenye Yang
University of Science and Technology of China
Division of Life Sciences and Medicine
443 Huangshan Rd
Hefei 230027
China

Dear Dr. Yang,

I am very pleased to accept your manuscript for publication in the next available issue of EMBO reports. Thank you for your contribution to our journal.

Yours sincerely,
